# Photoredox phase engineering of transition metal dichalcogenides

Juhwan Lim[1,2], Jung-In Lee[2], Ye Wang[2], Nicolas Gauriot[1], Ebin Sebastian[1], Manish Chhowalla[2], Christoph Schnedermann[1✉] & Akshay Rao[1✉]

Crystallographic phase engineering plays an important part in the precise control of the physical and electronic properties of materials. In two-dimensional transition metal dichalcogenides (2D TMDs), phase engineering using chemical lithiation with the organometallization agent *n*-butyllithium (*n*-BuLi), to convert the semiconducting 2H (trigonal) to the metallic 1T (octahedral) phase, has been widely explored for applications in areas such as transistors, catalysis and batteries[1–15]. Although this chemical phase engineering can be performed at ambient temperatures and pressures, the underlying mechanisms are poorly understood, and the use of *n*-BuLi raises notable safety concerns. Here we optically visualize the archetypical phase transition from the 2H to the 1T phase in mono- and bilayer 2D TMDs and discover that this reaction can be accelerated by up to six orders of magnitude using low-power illumination at 455 nm. We identify that the above-gap illumination improves the rate-limiting charge-transfer kinetics through a photoredox process. We use this method to achieve rapid and high-quality phase engineering of TMDs and demonstrate that this methodology can be harnessed to inscribe arbitrary phase patterns with diffraction-limited edge resolution into few-layer TMDs. Finally, we replace pyrophoric *n*-BuLi with safer polycyclic aromatic organolithiation agents and show that their performance exceeds that of *n*-BuLi as a phase transition agent. Our work opens opportunities for exploring the in situ characterization of electrochemical processes and paves the way for sustainably scaling up materials and devices by photoredox phase engineering.

Transition metal dichalcogenides (TMDs) exhibit a wide range of electronic properties based on their crystallographic structure[1,2]. Of particular interest is the phase transition from the trigonal 2H to the octahedral 1T (including 1T and distorted 1T) phase, which changes the electronic behaviour from a semiconducting to a quasi-metallic[1–3] state. This phase transition has been exploited to enhance field effect transistor device performances by lowering the contact resistance[3–5], to increase the hydrogen evolution reactivity[6–9] and to unlock a wide range of chemical exfoliations and functionalization down to single-layer TMDs[10–13]. More recently, it has also been used to advance flexible rectifying antennas[14] and lithium–sulfur battery technology[15].

Chemical lithiation using the organolithiation agent *n*-butyllithium (*n*-BuLi), which was first proposed in 1975 (ref. 16), remains the most popular route to achieve the 2H to 1T/1T′ phase transition. Here, the semiconducting 2H-TMD is immersed in *n*-BuLi, which yields the 1T/1T′ phase after several days. Mechanistically, *n*-BuLi first donates an electron to the 2H-TMD, which enables Li-ion intercalation into the van der Waals (vdW) gap to complete the 2H to 1T/1T′ phase transition[1,16]. This charge-transfer-limited reaction mechanism is further supported by electrochemical and electron microscopy studies, which suggest a sequential process of lithium adsorption through initial charge transfer,

followed by the phase transition and irreversible phase conversion at lower potentials[17–25]. However, we still lack a fundamental real-time nanoscale picture of the 2H to 1T/1T′ phase transition. This hampers the use of phase engineering of TMDs in crucial aspects: (1) the current reaction kinetics are extremely slow, requiring, for example, tens of hours for complete conversion of 2H-MoS₂ to the 1T/1T′ phase; and (2) the use of pyrophoric *n*-BuLi poses a substantial safety risk for any scale-up of these materials.

Here, using in situ optical reflectance interferometric contrast microscopy (RICM), we uncover mechanistic details of the 2H to 1T/1T′ phase transition with diffraction-limited resolution (about 200 nm). We discover a photoredox-activated reaction pathway that overcomes the otherwise slow charge-transfer-limited reaction occurring under 'dark' conditions, allowing us to markedly speed up the phase transition and demonstrate a new photoredox phase patterning methodology. Finally, with the obtained insights, we can eliminate the use of pyrophoric *n*-BuLi in favour of non-pyrophoric polycyclic aromatic hydrocarbon lithiation systems.

Figure 1a shows the experimental setup (left) used to monitor in situ the chemically induced phase transition from the semiconducting 2H to the quasi-metallic 1T(1T/1T′) phase (right). Mechanically exfoliated

[1]Cavendish Laboratory, University of Cambridge, Cambridge, UK. [2]Department of Materials Science and Metallurgy, University of Cambridge, Cambridge, UK. ✉e-mail: cs2002@cam.ac.uk; ar525@cam.ac.uk

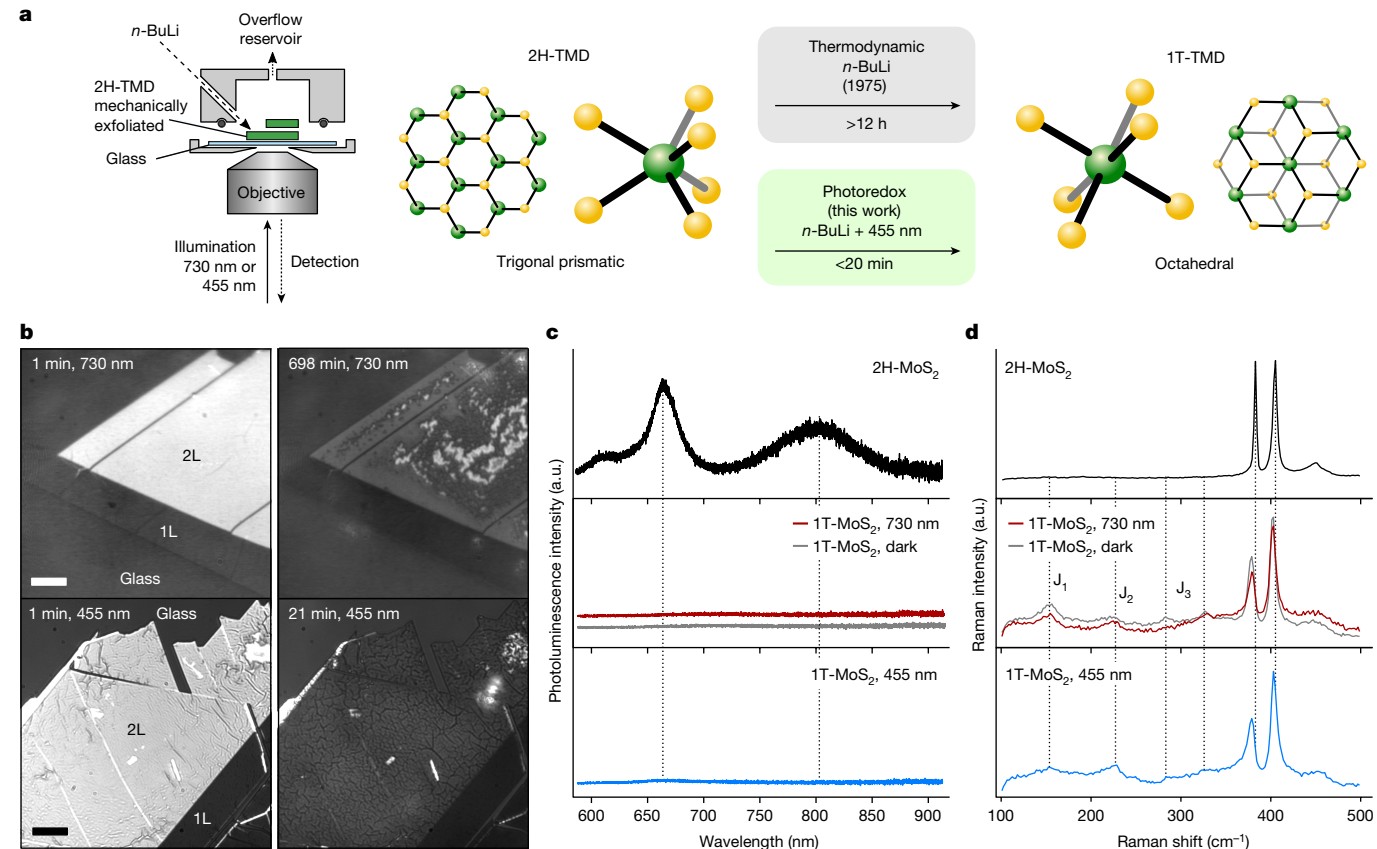

**Fig. 1 | Light-driven phase change of MoS₂ during chemical lithiation.**
**a**, Optical setup for this study and the schematic for light-driven 2H to 1T phase transition of TMD. Illumination with 455 nm accelerates the phase transition by up to two orders of magnitude. The bottom side of the flake is observed through a glass substrate, whereas the top side is in contact with the solution. **b**, Image of mono- (1L) and bilayer (2L) 2H-MoS₂ and 1T-Li$_x$MoS₂ under different illumination wavelengths. The 2H phase (left images) changed to 1T(1T/1T') phase (right) in 698 min under illumination with 730 nm (red, top row), and 21 min with 455 nm (blue, bottom row). **c,d**, Photoluminescence and Raman spectra of 2L 2H-MoS₂ (as exfoliated) and 1T-MoS₂ produced using three different methods: 48 h $n$-BuLi immersion in the dark (grey), 24 h of 730 nm (below band gap) illumination (red) and 25 min of 455 nm (above band gap) illumination (blue). The higher optical resolution achieved at 455 nm shows more fine structure in the flake (see Supplementary Fig. 10 for a comparison of the same flake under 455 nm and 730 nm). Scale bar, 5 µm (**b**).

2H-TMD flakes were placed in an air-tight, optically accessible cell that could be filled with $n$-BuLi using a syringe pump. The reaction was spatio-temporally resolved with an RICM using LED illumination at 730 nm and 455 nm.

Typical RICM images for mono- and bilayer MoS₂ before and after the phase transition under both illumination conditions are shown in Fig. 1b. We note that the higher diffraction-limited resolution at 455 nm shows a more detailed morphology compared with that at 730 nm (for more information, see Supplementary Information section 1). Independent of the illumination, at the start of the reaction the bilayer (2L) appeared brightly reflective, whereas the monolayer (1L) was substantially darker with intensity values similar to that of the glass substrate. The brightness reduces during the reaction and the images after 698 min under 730 nm illumination show a drastic reduction in the intensity, leaving behind a smooth surface at the edges of the flake and a more rugged central region characterized by irregular brighter and darker spots. We propose that the darkening is associated with the 2H to 1T phase transition. Flake darkening was also observed under 455 nm, but it took only 21 min, corresponding to a 33-fold acceleration of the reaction speed under 455 nm illumination.

We confirmed that the darkened 2L regions correspond to the 1T-MoS₂ phase by steady-state photoluminescence, Raman and reflectance microscopy measurements. Before chemical lithiation, bilayer 2H-MoS₂ exhibited photoluminescence at around 660 nm (direct bandgap) and around 800 nm (indirect bandgap) alongside pronounced Raman peaks at approximately 385 cm⁻¹ and 407 cm⁻¹ (E$_{2g}$ and A$_{1g}$ modes, respectively; Fig. 1c,d, black line). After the 2L fully darkened, we observed a loss of all photoluminescence emission, as well as red shifts of the E$_{2g}$ Raman peaks and the emergence of J-peaks in both illumination conditions, indicating successful 1T formation (Fig. 1c,d; 730 nm (red) and 455 nm (blue))[26]. We verified that the same photoluminescence and Raman signatures are observed when exposing 2H-MoS₂ to $n$-BuLi for 48 h under dark conditions (Fig. 1c,d, dark grey) and that the monolayer region is also successfully converted to 1T (refs. 1,26,27) (for more details, see Supplementary Information sections 2 and 3). Moreover, reflectance measurements show the loss of the A- and B-exciton absorption peak in line with the formation of a 1T phase[28,29] (for more sample characterization details, see Supplementary Information section 4).

These results confirm that the darkening of the flake observed in RICM corresponds to the phase transition from 2H-MoS₂ to its 1T phase and can be sensitively and optically monitored in situ. Furthermore, using below-bandgap illumination at 730 nm enables us to capture the phase transition in MoS₂ as it would occur under nominally dark conditions (Extended Data Fig. 1). By contrast, the above-bandgap excitation at 455 nm coincides with the C-exciton absorption band of 2H-MoS₂, suggesting a photo-activated mechanism that markedly accelerates the phase transition[30].

We study the mechanistic origin of the phase transition under 730 nm illumination by analysing the RICM images as a function of

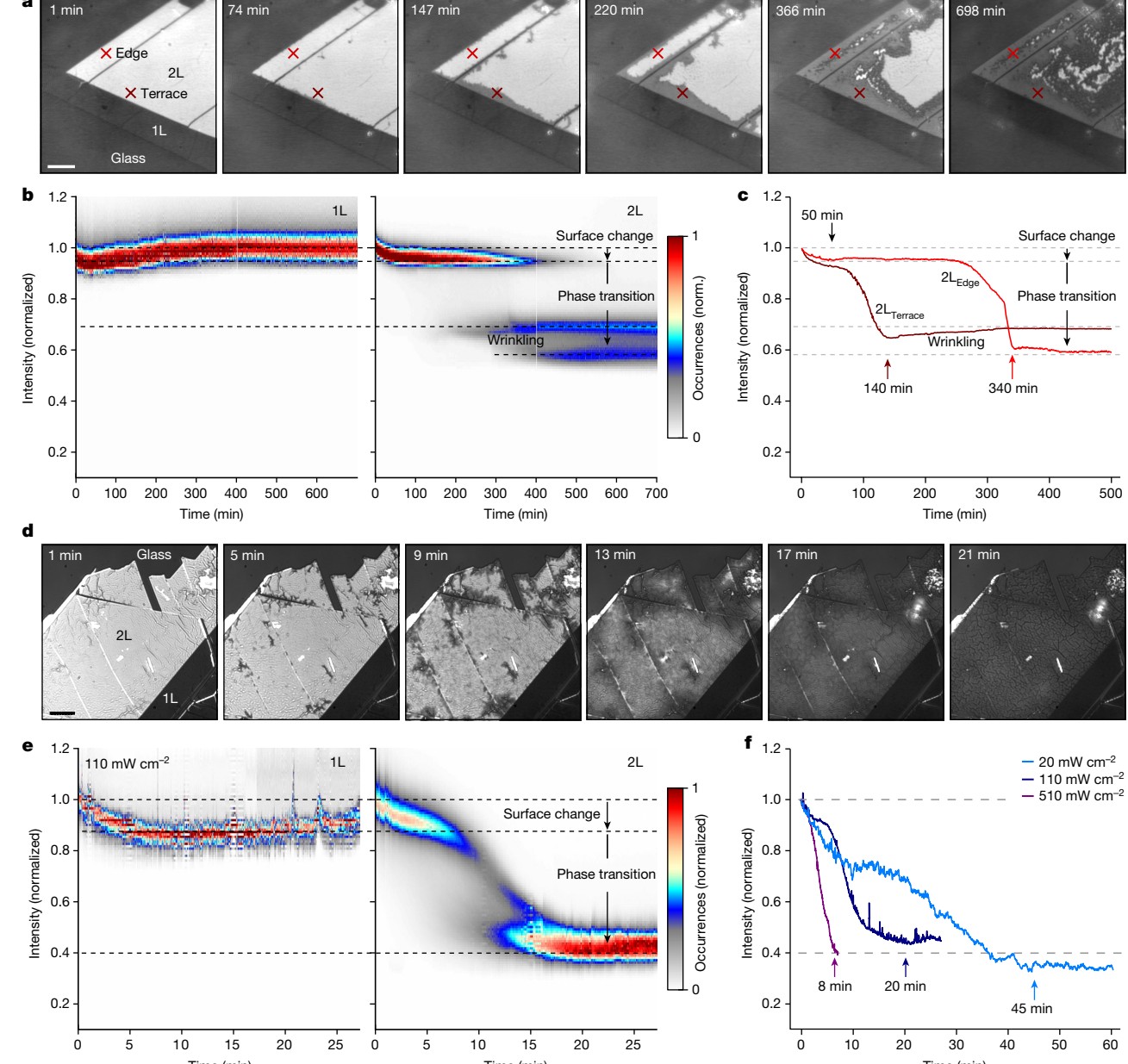

**Fig. 2 | Mechanistic explanation of light-driven phase transition of MoS₂ during chemical lithiation. a**, Snapshots of the phase transition visualized at 730 nm. **b**, Histogram analysis of the 1L (left) and 2L (right) regions. **c**, Selected reaction dynamics from two different points of the 2L indicated in **a**. Both spots are located at a similar distance (1 μm) from the edge. **d**, Snapshots of the phase transition visualized at 455 nm. **e**, Histogram analysis of the 1L (left) and 2L (right) regions. **f**, Power dependence of the reaction dynamics under 455 nm laser illumination for three different flakes. Each curve was obtained 1 μm away from the 2L/glass edge of the studied sample. Histograms are normalized to the mean of the intensity distribution 1 min after *n*-BuLi was added. Scale bar, 5 μm (**a**,**d**).

time after *n*-BuLi addition (Fig. 2a). After 74 min, the 2L edges become visibly darker, and by 147 min, a progression of a phase front at the 2L/1L terrace can be seen. A similar phase front progression at the 2L/glass edge was reached after 220 min. Once the 1T phase nucleated, the 1T/2H phase boundary moved into the flake with a constant velocity of 0.1 nm s⁻¹, remaining sharp throughout. Interestingly, after 366 min, brighter and darker regions formed whenever two-phase boundaries of different orientations converged. We attribute these regions to microscopic wrinkles arising from a lattice mismatch[20,31]. Similar phase frustration effects were previously observed for phase transitions occurring in lithium cobalt oxide[32]. After the wrinkles formed for the first time, the surface morphology of the 1T phase was no longer smooth but instead stayed rugged with brighter and darker features

emerging throughout the remaining phase transition (698 min, see also Supplementary Video 1; for more details, see Supplementary Information section 1).

To quantify the reaction dynamics at 730 nm, we performed a histogram analysis of the 1 L and 2 L regions. As shown in Fig. 2b (left), the 1L exhibited a single-peaked distribution in the histogram map. This feature quickly reduced in intensity to 0.9 over 50 min with no changes to its width, indicating a homogeneous surface change during the initial reaction with *n*-BuLi. As the 1L cannot intercalate Li ions, this initial decay suggests a facile electron donation to the surface. We note that the intensity at the glass/*n*-BuLi interface changes up to 15% over 12 h in this experiment, which accounts for the long-term increase in reflection intensity (Supplementary Information section 1).

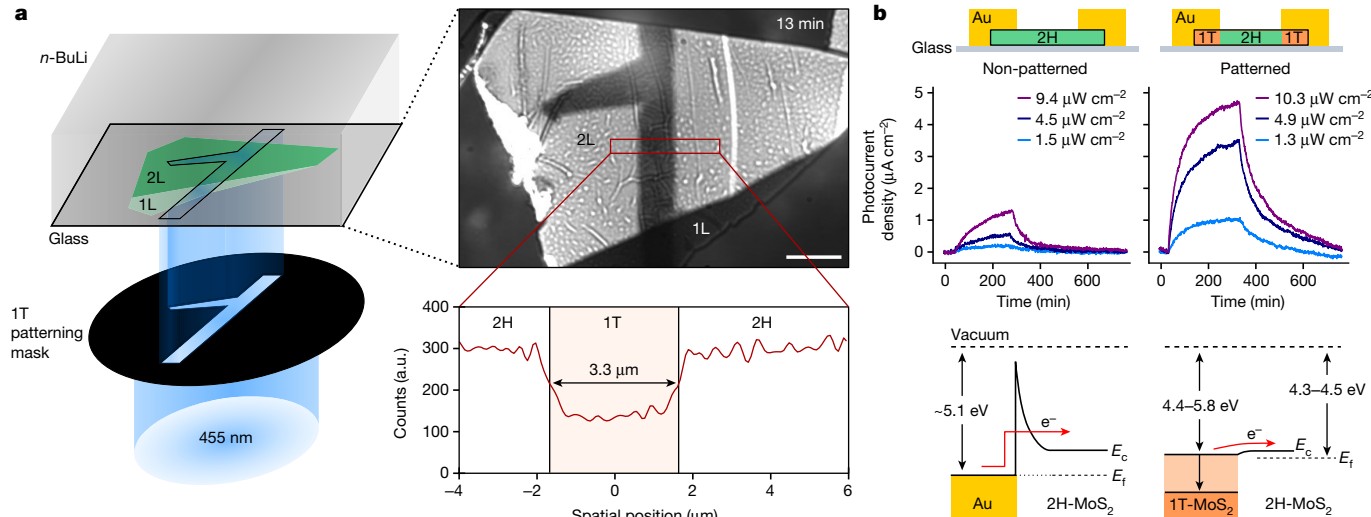

**Fig. 3 | Photoredox phase patterning on MoS₂. a**, Schematic of the photoredox phase patterning of a 1T phase pattern on 1L- and 2L-MoS₂ (left). The image of the patterned flake (right) and the average across the red rectangle show the profile of the patterned 1T phase. **b**, Response comparison of non-patterned (2H) and photoredox-phase-patterned (1T/2H/1T) 1L-MoS₂ photodetector. The bottom panel shows the ohmic-like transport behaviour by photoredox phase patterning. Scale bar, 5 µm (**a**).

Similarly, at the beginning of the reaction, the 2L (Fig. 2b, right) exhibited a single-peaked feature in the histogram map that rapidly decayed to a value of 0.9 over 50 min—the same timescale observed for the 1L. To confirm that this reduction corresponds to a surface change of the 2L, we carried out the photoluminescence measurements that showed that the vdW interaction between each layer is weakened within around 50 min (for more details, see Supplementary Information section 7). As the width of the initial feature does not change during this time, we attribute this process to a homogeneous surface electron donation, as observed for the 1L. After 50 min, the histogram peak at 0.9 reduces in occurrences and is converted into a lower intensity feature at an intensity of 0.7, which becomes prominent after about 300 min. This feature signifies the transition of the 2H phase (bright) to the 1T phase (dark) at the edges of the 2L (compare with Fig. 2a). After around 400 min, a second peak emerges at a lower intensity (approximately 0.6), accompanied by a broadening of both lower intensity features, coinciding with the formation of a wrinkled 1T region at the centre of the 2L.

To further illustrate the intrinsic variability of the phase transition as observed at 730 nm, Fig. 2c shows the intensity profile over time at the 2L/1L terrace and 2L/glass edge (2L$_{Terrace}$ and 2L$_{Edge}$, respectively). The 2L$_{Terrace}$ completed the phase transition in just 140 min, whereas the equidistant 2L$_{Edge}$ required 340 min. This discrepancy probably originates from different activation barriers along these two interfaces, highlighting how our approach can be used to map out intercalation energy barriers in these materials measurements (for more details, see Supplementary Information section 6).

Taken together, the first step of the phase transition in MoS₂ is rapid surface electron donation from *n*-BuLi to the whole flake. In 1L-MoS₂, this is sufficient to form the 1T phase. In 2L-MoS₂, Li-ion intercalation causes the 1T phase to form at the edges of the flake with a sharp phase boundary that moves with a constant velocity into the centre of the flake, suggesting a charge-transfer limited reaction mechanism. Finally, once the two-phase boundaries of different orientations converge, a wrinkled 1T phase is formed.

We repeated the same analysis for the 455 nm illumination. As shown in Fig. 2d, after 5 min, we already observed the 2L edges darkening and forming a sharp phase boundary between the bright 2H and dark 1T phases. After 13 min, however, this phase boundary was more diffuse and the centre of the flake appears noticeably darker. After 17 min, no clear phase boundaries could be identified, and the flake appeared to darken homogeneously with no signs of wrinkle formation (Supplementary Video 2).

Analysis of the histogram maps in Fig. 2e (left) for the 1L showed an identical, albeit accelerated surface reaction compared with the 730 nm illumination. Again, we observed an initial homogeneous darkening of the whole 1L to an intensity of 0.9 in just 5 min, followed by no further changes. The 2L (Fig. 2e, right) mimicked this intensity reduction in the first 5 min after the addition of *n*-BuLi, confirming that the 455 nm illumination accelerates the surface electron donation step of the reaction. After about 5 min, the initial distribution reduces in occurrences and shifts in intensity to form a new distribution at 0.4 within 10 min, in a markedly different behaviour than what we observed at 730 nm.

Mechanistically, these observations suggest a photoredox-activated mechanism. Initially, photo-excitation into the C-exciton of MoS₂ at 455 nm must lower the activation barrier for electron donation, to explain the accelerated surface darkening[33]. We propose that C-exciton excitation causes efficient surface-hole generation, thereby remarkably increasing the electron transfer rate (as seen by the change time of the initial darkening around 100 min at 730 nm compared with about 5 min at 455 nm) in line with previous transient absorption reports[34]. The associated opening of the vdW gap then enables rapid Li-ion intercalation from the edges of the flake. Owing to the continued photo-induced hole formation, this process is no longer strictly charge-transfer limited, causing the phase boundary to broaden and to accelerate the overall reaction speed.

To further support this mechanistic picture, we carried out power-dependent measurements on 2L-MoS₂, varying the illumination fluence at 455 nm between 20 mW cm⁻², 110 mW cm⁻² and 510 mW cm⁻². Figure 2f shows the intensity curves for regions located 1 µm away from the 2L/glass edge. We find that the photoredox-activated phase transition depends on power. Lower fluences (20 mW cm⁻²; Fig. 2f, light blue) approach the shape of the intensity curves observed under 730 nm illumination (Supplementary Video 3), indicative of a charge-transfer-limited reaction composed of an initial surface change followed by a period of constant intensity before the phase boundary traverses over the region. By contrast, higher fluences (510 mW cm⁻²; Fig. 2f, purple), lack clear features of a phase boundary and simply reduce in intensity to a constant value in as little as 8 min (about 90-fold acceleration), demonstrating that the process is deviating from a strict charge-transfer-limited reaction profile.

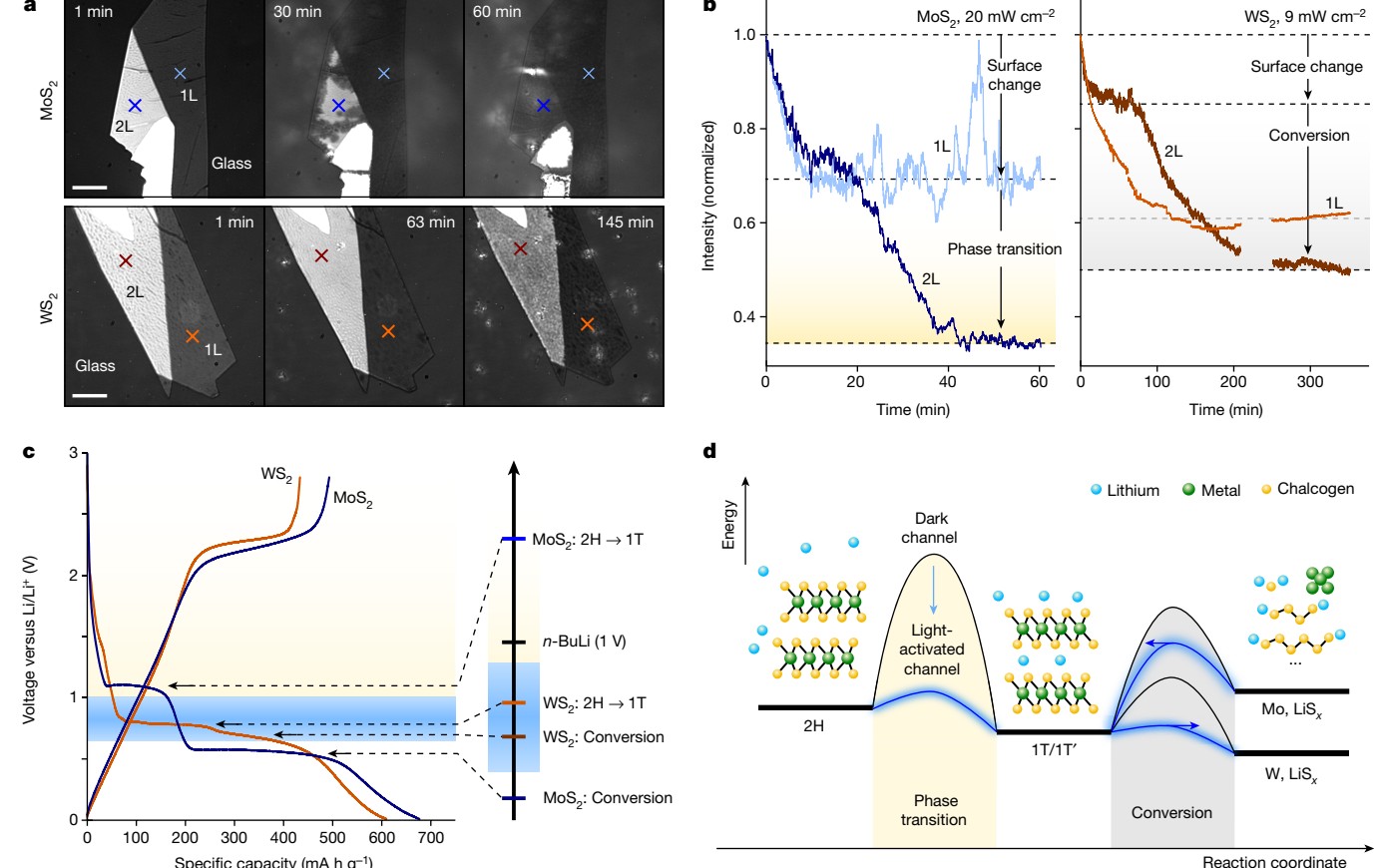

**Fig. 4 | Correlation of phase transition by chemical and electrochemical lithiation. a**, Chemically induced phase transition of $MoS_2$ and $WS_2$ observed in RICM. **b**, Reaction dynamics at selected points indicated in **a** for $MoS_2$ and $WS_2$ during chemical lithiation. No data were recorded for $WS_2$ between about 220 min and 250 min. **c**, Electrochemical discharge/charge voltage profiles of $MoS_2$ and $WS_2$ versus $Li/Li^+$. **d**, Energy landscape proposed for the phase transition and conversion reaction of $MoS_2$ and $WS_2$. The blue line shows the light-activated pathways. Arrows at the conversion barriers indicate the thermodynamic preference of the system. Scale bar, 5 μm (**a**).

A particularly promising application of the discovered photoredox-driven reaction pathway lies in the possibility of inscribing arbitrary phase patterns with diffraction-limited edge resolution into few-layer TMDs. Figure 3a shows the experimental setup used to create a well-defined T-shaped pattern of the 1T phase into 1L and 2L 2H-$MoS_2$. Using 455 nm illumination at a power density of 510 mW cm$^{-2}$, we were able to inscribe the desired phase pattern into 1L-$MoS_2$ in 30 s and into 2 L $MoS_2$ in 13 min (Fig. 3a, top right) and achieved a channel width of 3.3 μm with a deconvolved 1T/2H edge sharpness of $\sigma$ = 161 nm (Fig. 3a, bottom right). Photoluminescence and Raman imaging confirmed the successful phase transition (Extended Data Fig. 2 and Supplementary Fig. 9). It is worth noting that these power densities are about six orders of magnitude lower than conventional laser-writing approaches (for more details, see Supplementary Information section 8).

To demonstrate the electronic quality of the photoredox-patterned phase, we used our approach to inscribe a 1T/2H/1T phase junction into 1L 2H-$MoS_2$. Using a power density of 1 W cm$^{-2}$ enabled us to manufacture this junction in 10 s, without affecting the material. We then deposited gold electrodes onto the 1T phase area to fabricate a simple phase-engineered photodetector. As shown in Fig. 3b, the phase-engineered device exhibited enhanced photocurrent density to devices made with 2H-$MoS_2$, showing responsivity of 85 A W$^{-1}$ compared with 10 A W$^{-1}$ (ref. 35) (for more details, see Supplementary Information section 9). Using Fowler–Nordheim analysis, we verified that this enhanced photoresponsivity arises by switching from a Schottky contact behaviour to an ohmic-like contact behaviour (Fig. 3b, bottom), which enhances the transport properties[3,36] (Supplementary

Information section 10). Furthermore, it demonstrates that the diffraction-limited edge resolution does not notably impede electron mobility at the phase edges[35].

Next, we explored this photoredox reaction across different TMDs, such as $WS_2$, $MoSe_2$ and $WSe_2$. Here we will focus on the comparison of $MoS_2$ and $WS_2$, but a full discussion of all samples is provided in Supplementary Information sections 2, 12–14. Figure 4a compares the spatio-temporal changes of $MoS_2$ (top, blue) and $WS_2$ (bottom, orange) in the low-power regime on the addition of $n$-BuLi at 455 nm illumination. As described previously, the 2L-$MoS_2$ shows surface darkening together with phase boundary propagation, whereas the 1L-$MoS_2$ reduces only in intensity. Conversely, the 1L and 2L $WS_2$ proceed by a homogeneous surface darkening mechanism with no visible phase boundaries at any stage during the reaction (Supplementary Video 4).

To further analyse this difference, Fig. 4b shows the intensity curves for comparable 1L and 2L regions of the respective TMD. Although the 1L- and 2L-$MoS_2$ exhibit a simultaneous, rapid surface darkening followed by a decrease in intensity in 2L-$MoS_2$ because of the phase boundary motion, this behaviour is absent in $WS_2$. The 1L-$WS_2$ shows a rapid intensity decay within about 100 min, which is not matched by any dynamic feature observed in the 2L-$WS_2$. Instead, the 2L-$WS_2$ decays initially to an intermediate intensity not observed in the 1L-$WS_2$ and then continues to decay slowly over the next 300 min.

These results highlight that $WS_2$, in contrast to $MoS_2$, proceeds exclusively by a surface-driven reaction mechanism that seems different between 1L- and 2L-$WS_2$. We carried our Raman and photoluminescence measurements before and after the reaction of $WS_2$ with $n$-BuLi

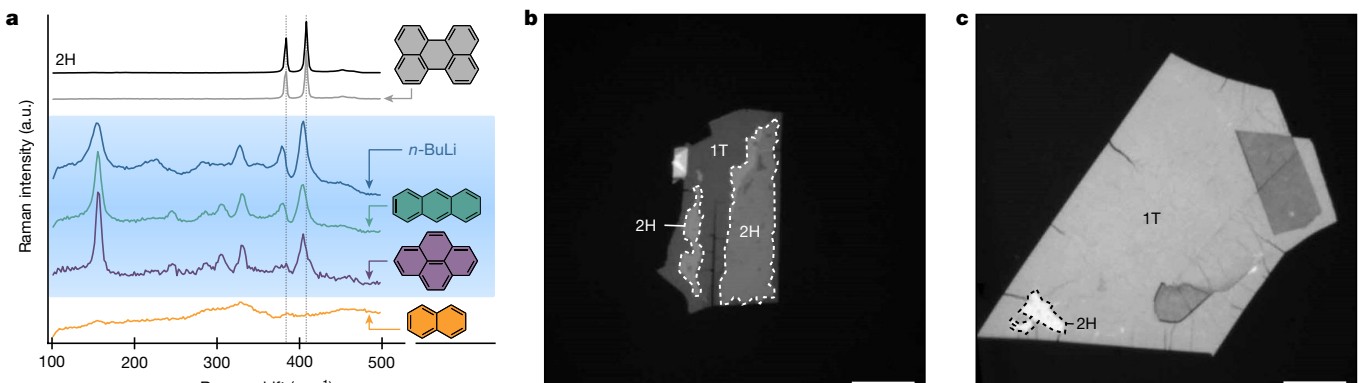

**Fig. 5 | Photoredox phase transition using PAH-Li system. a**, Raman spectra of PAH-Li treated MoS$_2$ with the chemical structure of PAHs. Blue-shaded region denotes redox-matched chemicals. Anthracene-Li (green) and pyrene-Li (purple) show the 1T Raman signature identical to the $n$-BuLi-treated sample (blue). **b**, Optical image of phase transition of thick MoS$_2$ by anthracene-Li treatment for 600 s in the dark. **c**, Optical image of phase transition of thick MoS$_2$ by anthracene-Li treatment for 10 s with 445 nm illumination. Scale bar, 10 μm (**b**,**c**).

(Supplementary Information sections 11 and 12). 2H-WS$_2$ showed the expected photoluminescence and Raman peaks. However, after exposure to 455 nm light in the presence of $n$-BuLi, the Raman spectrum indicated a 2H phase with a high signal-to-noise ratio, and no photoluminescence could be observed. On the basis of these results, we conclude that the chemical lithiation of WS$_2$ under 455 nm photo-excitation does not produce the desired 1T phase, but instead leads to phase conversion to tungsten and lithium sulfides[37]. We further attempted to resolve the phase transition in WS$_2$ under below-band-gap illumination at 730 nm; however, over the time frame of 48 h, no changes to the flakes could be observed, suggesting less favourable energetics for the chemical lithiation by $n$-BuLi, as compared with MoS$_2$.

To rationalize these findings, we fabricated electrochemical devices based on (multilayer) 2H phase TMD powders and measured the galvanostatic discharge–charge response of MoS$_2$ and WS$_2$ as shown in Fig. 4c (for more details, see Supplementary Information section 13). MoS$_2$ (blue) exhibited two plateaus during the galvanostatic discharge process at 1.1 V, corresponding to the reversible phase transition from 2H to 1T, and another at 0.57 V, assigned to conversion by irreversible decomposition[17,18]. As the redox potential of $n$-BuLi is 1 V (refs. 1,38), the charge-transfer-limited phase transition in MoS$_2$ proceeds by a small driving force of about 0.1 V. The slow reaction speed is, therefore, because of a reasonably large activation energy for electron donation. Illumination with 455 nm light increases the (surface) hole concentration and thereby reduces the activation barrier for electron donation, resulting in an accelerated reaction rate. Importantly, photo-excitation of MoS$_2$ does not provide enough driving force to activate the conversion reaction at 0.57 V.

This picture changes for WS$_2$, for which the phase transition and conversion potential are close together and located between 0.75 V and 0.8 V (Fig. 4c, orange). Chemical lithiation is, therefore, thermodynamically unfavourable, resulting in an extremely slow reaction with low yield after more than a week of $n$-BuLi soaking at room temperature (for more details, see Supplementary Information section 11). As for MoS$_2$, illumination at 455 nm reduces the activation barrier for electron transfer. However, owing to the close proximity of the phase transition and conversion, continuous photo-excitation cannot selectively stop at the 1T phase for WS$_2$. Instead, the conversion reaction is also photo-activated and occurs shortly after the 1T phase has formed. Examination of the MoSe$_2$ and WSe$_2$ corroborates this picture, in which MoSe$_2$ could be successfully transitioned to the 1T phase, and WSe$_2$ showed conversion behaviour (Supplementary Information sections 12–14).

Taken together, we propose the energy level scheme presented in Fig. 4d to explain our findings. For MoX$_2$ (X = S, Se), the thermodynamic landscape puts the reversible lithiated 1T phase into a local minimum with respect to its 2H phase and decomposition products[39–42] (for more details on reversibility, see Supplementary Information section 15). Although the activation barrier for the 2H–1 T phase transition is large, photo-excitation at 455 nm effectively reduces this barrier, leading to fast electron donation and Li-ion intercalation. In WX$_2$, the initial electron donation reaction is thermodynamically unfavourable and can be initiated only under 455 nm illumination. However, as the energy difference between 1T and decomposition products is very small, continuous illumination pushes the system too far, preventing controlled 1T formation, and ultimately resulting in conversion.

Finally, we sought to improve the sustainability of TMD phase engineering by removing the toxic and pyrophoric $n$-BuLi from the reaction. By considering the requirements of the reaction on the electrochemical potential, the chemical redox potential and illumination conditions, we developed several safe organo-lithiation agents based on polycyclic aromatic hydrocarbons (PAHs) and lithium metal in tetrahydrofuran solvent. These chemicals furnish a solution of pi-radical anions counterbalanced by Li cations with sufficient driving force for electron donation to TMDs, offering a route towards replacing $n$-BuLi.

Figure 5a presents the Raman spectra of (multilayer) MoS$_2$ treated with different PAH-Li agents. Anthracene-Li (AnLi, green, 0.91–1.03 V) and pyrene-Li (purple, 0.73–0.93 V), fall into the acceptable redox potential range between 0.57 V and 1.1 V and, therefore, successfully convert 2H-MoS$_2$ to its 1T phase, as evidenced by their close spectral match to $n$-BuLi (blue). By contrast, perylene-Li (grey, 1.19–1.35 V) did not react with the 2H phase because of insufficient redox potential, whereas naphthalene-Li (orange, 0.26–0.46 V) resulted in undesirable decomposition because of a high driving force[43].

To explore the effect of changing the redox reagent on the photoredox-driven phase transition dynamics, we investigated exfoliated, thick MoS$_2$ flakes on glass and added AnLi. Figure 5b shows an ex situ optical image of a thick flake under 'dark' conditions after 10 min of reaction time, showing a partial phase transformation. Noticeably, the extracted phase front speed of about 6 nm s$^{-1}$ with AnLi in a thick flake was remarkedly faster than with $n$-BuLi in a bilayer (0.08 nm s$^{-1}$), showcasing the effect of the higher driving force provided by AnLi. On above-gap illumination (Fig. 5c; 10 s illumination at 130 mW cm$^{-2}$), the reaction is yet again substantially accelerated with an estimated phase front speed of up to 5 μm s$^{-1}$, about six orders of magnitude faster than under dark conditions for a bilayer. Taken together, our results show that even thick TMD flakes can be readily transformed by combining the effect of redox potential matching with above-gap illumination. Moreover, the AnLi system also worked for MoSe$_2$ and WS$_2$ by accelerating the lithiation reaction (Supplementary Information sections 16 and 17).

In summary, our study provides mechanistic insights into the 2H to 1T phase transition in few-layer TMDs by real-time visualization of the chemical lithiation reaction. We develop a photo-activated reaction pathway that accelerates the phase transition reaction, allowing for rapid phase-patterning of $MoS_2$. Finally, we use these insights to replace pyrophoric *n*-BuLi from the reaction with safe organolithiation agents (PAH-Li). These reagents make the process safer and greener and also reduce reaction times by six orders of magnitude compared with conventional *n*-BuLi treatment.

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

## Methods

### Materials

$MoS_2$, $MoSe_2$, $WS_2$ and $WSe_2$ (HQ graphene) were exfoliated on a microscopic cover glass substrate (thickness = $170 \pm 5$ μm) by mechanical exfoliation using a polydimethylsiloxane (PDMS) tape.

### Cell design and chemical lithiation process

The air-tight cell for this study was designed in two parts: as a chemical reservoir and a base plate with an optical window. The reservoir had a volume of about 2 ml, which was enough to monitor slow reaction dynamics for a few days avoiding any issues from evaporation. The two ports of the reservoir part were assembled with tubes, which were connected to syringes. The first syringe was prepared by filling approximately 3 ml of $n$-BuLi (1.6 M in hexanes, Sigma-Aldrich), and the second empty syringe was prepared as a container for exhaust air (nitrogen) when injecting the chemical inside the cell. The glass substrate with TMD flake, two parts of the cell with two syringes and an O-ring (diameter 1.8 mm, EL-CELL) were assembled inside the nitrogen glovebox. The cell was placed on a custom-built RICM. After locating the desired flake, we injected $n$-BuLi for around 1 min, resulting in partial filling of syringe 2. The top side of the exfoliated flake is in contact with $n$-BuLi. The bottom side contacts the glass substrate through which the flake is observed during the RICM measurements. The $n$-BuLi level in both syringes was carefully monitored to ensure the flake remained fully immersed during the experiment.

We note that it has been reported that both distorted 1T′ phase and 1T phase coexist[2,25,28,29,42,44]. However, the two phases are indistinguishable at the optical resolution, and we will denote the 1T′/1T phase as 1T throughout this study. Lithiation of 2H-TMDs forms 1T-$Li_x$TMDs, which are stable in air[2,25]. To remove the excess butyllithium and other organic by-products[44], we first washed the sample using an excess amount of $n$-hexane. The washing step was done inside the cell by injecting the $n$-hexane using a syringe to avoid air exposure and minimize the chance of possible oxidation of surfaces. After steady-state optical characterization (photoluminescence and Raman), we rinsed the sample again using an excess amount of distilled water (DI), acetone and isopropyl alcohol (IPA) to remove all lithium (1T-TMDs).

### Steady-state optical characterization

The microscope steady-state photoluminescence and Raman measurements were carried out under ambient conditions using a Renishaw inVia Raman microscope using the excitation laser sources of 532 nm. Before all measurements, the spectrometer was calibrated using a silicon reference sample for correcting the instrument responses. The laser was set to a power of 0.05% (<0.5 μW) focused on the designated point of the flake using a 100× long working distance objective (numerical aperture = 0.85), and the emission was collected in streamline mode and dispersed by 1,800 l mm$^{-1}$ grating.

Reflection microscopy was performed with a custom-built microscope. A variable wavelength excitation was provided by a pulsed super-continuum white light source (Fianium Whitelaser) coupled to a monochromator (Bentham TMC 300). The sample was imaged, at each wavelength, in reflection geometry onto an EMCCD camera (Photometrics QuantEM 512SC) with a 60× oil immersion objective (UPLFLN60XOI, Olympus). The microscopic photoluminescence image was taken on the same setup, using 532 nm excitation and imaging onto the EMCCD camera with a 660-nm bandpass filter (Thorlabs).

### In situ RICM measurement

**Methods.** RICM measurements were carried out using a custom-built microscope setup reported in detail previously[32,44]. Furthermore, the microscope was equipped with a 455-nm LED (M455L4, Thorlabs) coupled into a single-mode fibre. The output of the fibre was collimated with a condenser lens and imaged into the back focal plane of the objective (UPLXAPO100XO, Olympus) to achieve wide-field illumination. The 730 nm illumination was carried out by inserting a 730-nm bandpass filter (FB730-10, Thorlabs) into the illumination path of a SOLIS-740C (Thorlabs). The magnification of the microscope was 167× and confirmed with a resolution target. A 2 × 2 software-based pixel binning, leading to a pixel pitch of approximately 70 nm per pixel, and, depending on the experiment, 10–30 frame binning was applied to improve the image quality. The effective acquisition frame rates varied between 0.05 Hz (730 nm illumination) and 3 Hz (455 nm illumination, 510 mW cm$^{-2}$) ensuring sufficient time resolution to monitor the phase transformation dynamics.

Throughout the experiment, the sample was kept in focus using a hardware-based autofocus routine[32]. No images were recorded during the injection of $n$-BuLi because of the large pressure fluctuations and associated focus changes of the sample. For the patterning experiments (Fig. 3a), a shadow mask was placed into the illumination path of the microscope such that it was imaged onto the sample. The flake was carefully positioned with respect to the mask in the absence of $n$-BuLi. Subsequently, the illumination was turned off and $n$-BuLi was injected as before. After 1 min, the syringe pump was turned off and the 455 nm illumination was switched on to initiate the photo-driven phase transition. The light source was turned off once the intensity of the flake was reduced to the 1T phase value, and the sample was washed immediately afterwards.

**Analysis.** Recorded image stacks were corrected for sample drift as previously described[32]. No other post-processing was applied.

For the histogram analysis presented in Fig. 2, the image stack was masked to include only 1L or 2L regions. The first image in each stack served as a reference frame to compute an average intensity value for each masked region. For each image in the stack, each pixel value in the masked image was subsequently divided by this reference intensity value to generate an intensity distribution centred at 1 (the mean intensity of the first image is computed and used to normalize all subsequent intensity histograms). For each image in the stack, a histogram with 120 equally spaced bins ranging from intensity values of 0.1–1.2 was then computed, yielding the histogram plots shown in Fig. 2. Generally, no normalization for the illumination intensity was required because of the high stability of the LED light sources (<5% power drop over 5 days continuous illumination). The intensity histogram is in the first instance a convolution of the actual intensity value and the noise characteristics of the camera. We verified shot-noise limited performance and can conclude that the noise broadening is well below 1%, causing no notable broadening of the normalized histograms.

During long-term experiments (>1 h), necessary to capture the phase transition dynamics under 730 nm illumination, we observed noticeable changes to the intensity at the glass/$n$-BuLi interface of up to 15% over 12 h. These changes match the observed long-term increase observed in Fig. 2b,c and must therefore be correlated. Given that our light sources are stable, the origin of this behaviour must be related to either a change in the refractive index of the $n$-BuLi solution or an increased out-of-focus scattering mechanism.

Apart from this slow intensity increase, the long-term imaging at 730 nm shows several landing events of several cluster-like particles. These particles are probably formed in solution reactions involving the highly reactive $n$-BuLi. We, therefore, believe that the long-term increase in reflection intensity is most likely associated with out-of-focus scattering, which our wide-field microscope will be sensitive to.

**Electrochemical measurement.** The electrochemical performance of bulk transition metal dichalcogenides as working electrodes was conducted in a coin 2032-type cell. The working electrodes were fabricated by casting a slurry of 80% active material, 10 wt% conducting agents (super-P) and 10 wt% binder (polyvinylidene fluoride) in $N$-methyl-2-pyrrolidinone on copper foil. The coated electrodes were

then dried at 50 °C in a vacuum overnight. The test cells were assembled in an argon-filled glove box (<0.5 ppm of oxygen and water) with lithium chips cut to a round shape (diameter 16 mm) as a counter electrode, a porous polypropylene separator (Celgard 2400) and 1 M LiPF6 in ethylene carbonate/diethyl carbonate (v/v) by adding 5 wt% fluoroethylene carbonate as the electrolyte. The mass loading of the active material was approximately $1–1.5$ mg cm$^{-2}$. After ageing at room temperature for 20 h, the cells were ready for detailed electrochemical tests. Galvanostatic charge–discharge tests were performed on a LAND Battery Tester at the current density rate of 0.05 C according to the theoretical capacity of each active material within the voltage range of $0.01–2.8$ V. Cyclic voltammetry at a rate of 0.1 mV s$^{-1}$ was conducted on the potentiostat instrument of Biologic in the potential range of $0.01–2.80$ V versus Li$^+$/Li.

**Photodetector fabrication and measurement.** The photodetector was fabricated by metal deposition using a shadow mask. A gold pad (50 nm) was deposited selectively on the 1T/1T′ region using electron-beam evaporation (PVD 200 Pro, Kurt J. Lesker). Photodetector measurement was conducted by illuminating a monochrome laser (MCLS1, Thorlabs) through a customized lens (Mitutoyo) and recording the photocurrent using a semiconductor analyser (Keithley 4200).

## Data availability

The data underlying all figures in the main text are publicly available from the University of Cambridge repository (https://doi.org/10.17863/CAM.110010).

## Code availability

All codes used in this work are available from the corresponding authors upon reasonable request.

44. Merryweather, A. J. et al. Operando monitoring of single-particle kinetic state-of-charge heterogeneities and cracking in high-rate Li-ion anodes. *Nat. Mater.* **21**, 1306–1313 (2022).

**Acknowledgements** We thank P. Knight and the members of the workshop at the Department of Material Science and Metallurgy at the University of Cambridge for their technical support for experimental design and cell fabrication. This project received funding from the European Research Council under the Horizon 2020 research and innovation programme of the European Union (grant agreement no. 758826 (SOLARX) to A.R.). This work was supported by the Engineering and Physical Sciences Research Council (grant EP/W017091/1). E.S. received funding from the UKRI postdoctoral individual fellowship (grant reference no. EP/Y026659/1). A.R., C.S. and J.L. acknowledge support from the Faraday Institution Degradation Project. J.-I.L., Y.W. and M.C. acknowledge support from the Faraday Institution LiSTAR programme and characterization project (EP/S003053/1, FIRG014 and FIRG012), the NEXGENNA programme (FIRG018), the Royal Society Wolfson Merit Award (WRM\FT\180009), and the European Research Council (ERC) Advanced Grant under the Horizon 2020 research and innovation programme of the European Union (grant agreement no. GA 101019828-2D-LOTTO). This work was funded by the UKRI. For open access, the author has applied a Creative Commons Attribution (CC BY) licence to any Author Accepted Manuscript version arising.

**Author contributions** A.R. conceived the project. C.S. and J.L. developed the project, designed and built the experiments and performed the RICM measurements, including the above-gap photoredox phase engineering. C.S. analysed the RICM data. J.L. and N.G. performed the steady-state optical characterization and photodetector fabrication. Y.W. and J.L. conducted the photodetector measurements. Y.W. performed the Fowler–Nordheim analysis. J.-I.L. performed the electrochemical measurement and analysis. J.L., C.S. and E.S. synthesized the PAH-Li system. M.C. and A.R. supervised the work.

**Competing interests** J.L., C.S. and A.R. have filed a patent application based on some aspects of this work.

**Additional information**
**Correspondence and requests for materials** should be addressed to Christoph Schnedermann or Akshay Rao.

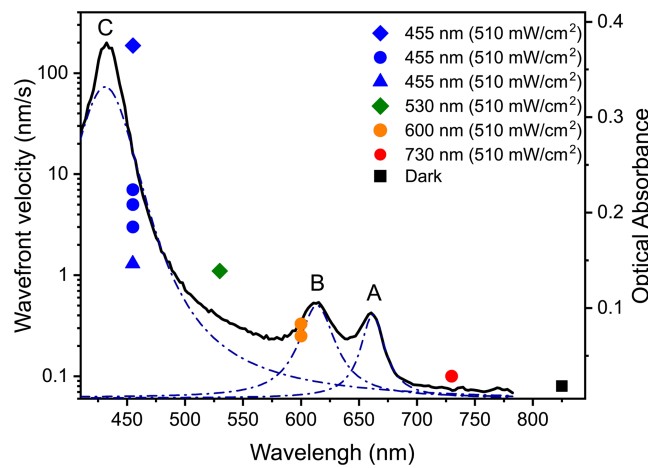

**Extended Data Fig. 1 | Chemical phase transition speed with respect to the illumination wavelength and power, evaluated by wavefront velocity of bilayer.** The wavefront velocity of each illumination follows left y-axis (nm/s) and black line is optical absorbance (right y-axis) (more information in Supplementary Information 5).

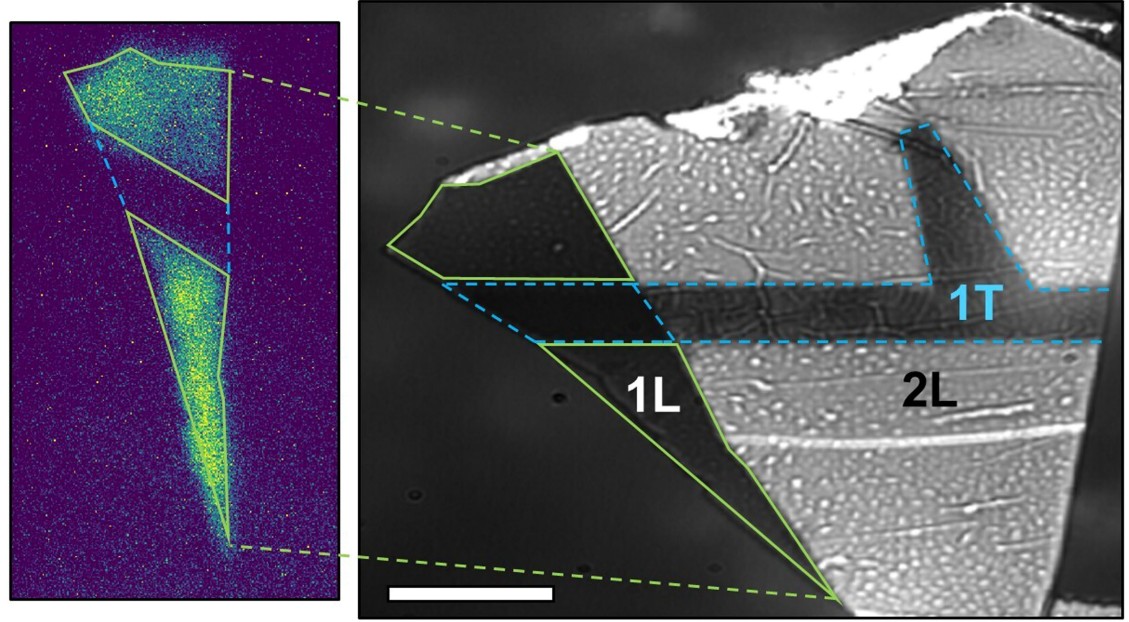

**Extended Data Fig. 2 | PL imaging of photo-redox phase patterned MoS$_2$.** MoS$_2$ flake in Fig. 4a, Scale bar, 5 μm.