## [Peer Review File · Nature]

Manuscript Title: Photo-Redox Phase Engineering of Transition Metal Dichalcogenides

Reviewer Comments & Author Rebuttals

Reviewer Reports on the Initial Version:

Referee #1 (Remarks to the Author):

The manuscript reported a phase transition of TMD semiconductors by using photo-accelerated chemical lithiation method. In general, using Li ion intercalation approach to realize phase transition has been widely studied and utilized for TMD-based electronic device applications and performance improvement. Although the work demonstrated in this manuscript shows the direct observation of the phase engineering during the intercalation process, the lack of supporting evidence as well as scientific novelty makes it under the standard of Nature publication in the present form. The following comments are hereby for further improvement of the manuscript:

First, more experimental details should be provided. Which side of the exfoliated flake was monitored? The top side in contact with n-BuLi or the bottom side contacting the glass substrate? In Fig. 1b, how was the 1L-MoS₂ and 2L-MoS₂ stacked?

The comparison between the microscope images of Fig. 1b is also dependent on the initial surface morphology of the flakes, like the different observation of smooth surface or rugged region with irregular brighter and darker spots. This weakens the comparison and conclusion about the illumination duration period.

The Raman peaks in Fig. 1d of both 2H and 1T phases have demonstrated strong and clear A_{1g} and E_{2g} peaks. However, such representative peaks are usually dampened after the phase transformation after n-BuLi treatment. Does this suggest incomplete phase transition? In addition, the J-peaks intensity of 455 nm illuminated sample is not as strong as the 730 nm illuminated sample. This should also be further clarified.

The authors claimed that there exists another pathway which proceeds via the surface penetration of Li ions in the center of the flake, according to Fig. 2e. However, how was such penetration formed, and is it random with more experimental evidence?

The authors mentioned that the phase conversion of WS₂ is irreversible. Then how was the phase transition of MoS₂? Is it reversible? Based on the discussion of the mechanism, it seems that such 2H-1T transition can be reversed. Experimental evidence is preferred to confirm if such reversion can be achieved.

What is the contact property between 1T-MoS₂ and Au electrode? The enhanced photocurrent cannot unambiguously confirm the lower contact resistance by forming the semi-metallic 1T phase. The Schottky barrier height can also affect the carrier transport which can be influenced due to different work function of the contact stack system.

One last comment concerning the practical application, the demonstrated approach utilizes optical illumination with different wavelength and power intensity, which can also introduce significant impact on the intrinsic property of the TMD materials as well as the TMD-based electronic devices. The authors are suggested to make a more comprehensive discussion regarding this issue.

Referee #2 (Remarks to the Author):

In this work, the authors report the use of an in situ optical reflectance microscopy technique to monitor the phase conversion of transition metal dichalcogenides from the 2H polytype to the 1T/1T' polytype via reaction with n-BuLi. They find that 455 nm illumination accelerates the polytype conversion considerably, and also enables photo-patterning of polytype domains. This is attributed to a photo-excitation mechanism, in which 455 nm illumination excites the C-exciton, which lowers the electron donation barrier and hence accelerates the polytype conversion. The study is quite detailed and provides a useful picture of the dynamics of polytype conversion under "normal"/dark and photo-accelerated conditions. The comparison of different TMDs increases its usefulness to the 2D materials community and reveals some intelligible trends. However, in my opinion, while technically strong, it seems the general impact of this manuscript at present is a bit lacking for a work to be published in Nature. Specifically, the authors do not yet demonstrate very distinctive benefits or scientific advances that result from the marked acceleration of the polytype conversion on its own or the photo-patterning concept (the photodetector demonstration is interesting, but it is not clear if the device performance is a major advance or if the approach improves fabrication/processing [speed, cost, etc] compared to other approaches). In addition, the claims of improvements in safety, scalability, and yield are somewhat difficult to understand, as n-BuLi must still be used.

In its current form, I think the manuscript is more suitable for a more specialized journal.

General question: How do the properties of the 1T polytype produced under 455 nm illumination and under "dark" conditions differ? The authors state that the conversion under 455 nm illumination results in fewer/no wrinkles. Since the greatest interest in polytype engineering stems from the promise of these materials in device applications, it would be useful to know how the transport properties of 1T-MoS₂ or 1T-MoSe₂ produced via these two different pathways compare, in order to evaluate the impact of the technique presented in this manuscript.

Figure 2, discussion of histogram analysis: Why are the intensity distributions after 1T conversion considerably sharper under 730 nm illumination compared to 455 nm illumination? Although only one intensity peak is observed under 455 nm illumination, could the broadness of this intensity suggest some inhomogeneity that the authors have not discussed?

Beginning on page 5: The discussion of the intensity values would be easier to follow if the authors specified more clearly in the main text that all of these values refer to normalized intensity, and what the normalization factor is.

Page 6, 2nd and 3rd full paragraphs: The authors state that "Li-ion intercalation through the surface layer of the 2L" is "not dependent on later Li-ion diffusion through vdW gap." What is the mechanism of this surface intercalation - do the authors think it occurs through defect sites? In addition, there appears to be a logical discrepancy with the statement at the bottom of the page about the power dependence study, where the authors state that the higher fluences demonstrate "that the process is governed primarily by a diffusion limited reaction profile." What is the nature of the diffusion process the authors refer to here?

Page 7, 1st paragraph: The statement "by accelerating the reaction, chemical safety risks associated with the pyrophoric nature of n-BuLi can be mitigated" seems rather misleading and unsubstantiated. In the "photo-redox" process proposed, n-BuLi must still be used. The risks associated with n-BuLi originate more from its handling than from the reaction time in question. If the amount or concentration of n-BuLi could be reduced through illumination, that might support such a claim. As is, the authors should modify this statement.

Page 9: The authors state the utility of the process described in patterning "few-layer" TMDs. The main examples in the paper are for 2L and 1L samples. One might expect that the lower surface area-to-volume ratio for increasing layer numbers would hamper the surface-mediated mechanism proposed for polytype conversion under above-bandgap illumination. In addition, the evolution of the bandgap from direct to indirect going from monolayer to multilayer flakes is not discussed. What, for example, is the effect of illumination and polytype conversion in 2L samples that are not connected to a 1L region? It would be helpful if the authors could comment on the utility of this approach for different thicknesses of MoS₂ or MoSe₂ flakes.

Referee #3 (Remarks to the Author):

The manuscript reports on an interesting study of photo-assisted 2H-1T phase transition in TMDCs using n-BuLi chemical lithiation. The work expands the current knowledge on the mentioned phase transition through the introduction of the technologically relevant blue illumination driven process, which can also be used for an efficient and simple patterning. The study is mostly conducted via in-situ optical reflectance experiments in a controlled environment. On the downside, even though the advancement in the field relates almost exclusively to the photo-assisted process, its explanation is vague and relies on an assumption which the authors do not try to prove (e.g. through the use of different illumination wavelengths that would probe other excitons, or, in contrast, be away from their energies). Similarly, the part on the differences between Mo and W-TMDCs could be strengthened by a reversibility study. Several other issues are listed below. To summarize, in spite of an undeniable appeal of the study to a large readership in the field of 2D materials, the limited novelty and depth of the work make the manuscript in the current form not well suited for the Nature journal.

Specific comments (also briefly including the ones mentioned above):

- 1) The explanation of the photo-assisted process is vague and should be strengthened at least by exploring the effects of other illumination wavelengths in/out resonance with (other) excitons. Ideally, this hypotheses should also be supported by theoretical calculations.
- 2) The reversibility of the phase transition in Mo-TMDCs could be shown by adjusting the lower cut-off potential in the CV, and, then, the material could be analysed ex situ.
- 3) One very important and confusing issue relates to the phase transition in 1L MoS₂. Page 3 states that the monolayer region is also successfully converted to 1T, but the reference points to Supplementary figures that do not seem to show anything like that (e.g., suppl. Fig. 5 shows a Raman spectrum only of a disordered 1L, with no clear signs of a 1T phase). The statement of the monolayer being converted to 1T is then made several other times, e.g., in relation to the photodetector. However, Figure 2 and its discussion clearly documents that the only process taking place on 1L is the surface charge, not the phase transition, which is shown only for 2L. This ambiguity should be thoroughly sorted out in the whole manuscript.
- 4) Concerning the proposed mechanism of the through-the-layer lithiation for WS₂, the authors should also consider and discuss the possibility that the lithiation directly leads also to an irreversible disorder connected with this process.
- 5) Page 5 states “This discrepancy likely originates from different activation barriers along these two interfaces,...”. The authors should expand on why should the activation barriers be different.
- 6) One could imagine that a further illumination of the patterned photodetector by blue light (even after the fabrication) might still lead to some mobilization of free Li ions into the photodetector area. Could the authors consider and discuss that possibility?

Author Rebuttals to Initial Comments:

Dear the reviewers,

We greatly appreciate time and interest in our manuscript. In this letter, we have addressed each response to the valuable comments from the reviewers.

In Part 1, we provide a point-by-point response to the scientific comments, experimental details, and analysis method requested by the reviewers.

In Part 2, we provide detailed explanations which are now part of the manuscript, on further studies we have conducted in response to the reviewers' suggestions:

- Section 1. Fowler-Nordheim (FN) Tunneling Plots.
- Section 2. RCM measurements using Different Wavelengths of Light.
- Section 3. Image analysis for structural dependent phase transition.
- Section 4. Reversibility of the Reaction.
- Section 5. Re-examination of Raman Data.
- Section 6. Redox-based organolithiation agent for novel chemical phase engineering: PHA-Li System.

Part 1. Response to reviewer's comments

Referee 1

The manuscript reported a phase transition of TMD semiconductors by using photo-accelerated chemical lithiation method. In general, using Li ion intercalation approach to realize phase transition has been widely studied and utilized for TMD-based electronic device applications and performance improvement. Although the work demonstrated in this manuscript shows the direct observation of the phase engineering during the intercalation process, the lack of supporting evidence as well as scientific novelty makes it under the standard of publication in the present form. The following comments are hereby for further improvement of the manuscript:

We thank the reviewer for their constructive questions and comments. Below, we have added supporting evidence related to each comment. We have carefully re-examined our Raman data in order to clarify the discussion on Raman signature of 1T MoS₂. We also now demonstrate explicitly that our photodetector exhibits direct-tunnelling behaviour, via use of a Fowler-Nordheim model. This is direct evidence of switching to direct tunnelling by phase engineering, which reduces the contact resistance and enhances the photodetector performance. Moreover, we have enhanced the clarity of our explanation regarding the experimental setup, underlying mechanisms, and the influence of optical excitation on the material itself and image resolution.

We would like to highlight that we have taken our study a step further by leveraging our findings to propose a safer and faster chemical route for rapid phase engineering, by replacing the hazardous and pyrophoric chemical, n-butyllithium, with a poly-aromatic hydrocarbon-Lithium systems which are safe, fast, and easy to use. With this new approach, the phase transition

for few-layer systems takes mere seconds and can be accomplished even in thick (i.e. bulk) MoS₂ within ~10 s. This result has been added in this letter (Part 2-section 6) and new added figure 5 in the manuscript.

Comment 1: First, more experimental details should be provided. Which side of the exfoliated flake was monitored? The top side in contact with n-BuLi or the bottom side contacting the glass substrate? In Fig. 1b, how was the 1L-MoS₂ and 2L-MoS₂ stacked?

The measurement focusses on the side of the flakes in contact with glass, within the inverted microscope configuration shown in Fig. 1a. All flakes have been mechanically exfoliated and optically characterized to determine the numbers of layers. So, 2L-MoS₂ is naturally stacked bi-layer MoS₂, which has been exfoliated on thin glass and characterized as bilayer with both PL and Raman spectra.

We have now clarified this aspect in the methods section (1.1 in SI) and figure caption. In Supplementary Information section 1.1., we added following explanation: The top side of the exfoliated flake is with contact with n-BuLi. The bottom side is contacting the glass substrate through which the flake is observed during the RICM measurements.

In caption for Fig. 1 A, we added following explanation : The bottom side of the flake is observed through a glass substrate, while the top side is in contact with the solution.

Comment 2 : The comparison between the microscope images of Fig. 1b is also dependent on the initial surface morphology of the flakes, like the different observation of smooth surface or rugged region with irregular brighter and darker spots. This weakens the comparison and conclusion about the illumination duration period.

We appreciate the concern the reviewer raises regarding Fig. 1b and apologise for not stating where this difference originates from. There are two main sources that affect to the observed roughness difference, the optical resolving power of the microscope and the exfoliation method.

The sample observed under blue light (455 nm) will appear sharper compared to red (730 nm) illumination due to the difference in diffraction limit, which can be estimated by the Abbe formula: $d = \lambda/(2NA)$. Here, d is the size of a feature that can be truthfully represented, λ is the wavelength and NA the numerical aperture of the objective (1.4 in our case). At 730 nm, $d_{730} = 260$ nm, while at 455 nm, $d_{455} = 162$ nm. A rough surface will therefore appear significantly smoother when inspected at 730 nm compared to 455 nm, since the optical image represents to first approximation a convolution of the actual surface roughness with a Gaussian point spread function of full width half max = d^1 .

To further quantify the initial heterogeneity, we have further analysed our experimental data and examined the width of the intensity histogram throughout the reaction (see also response to Reviewer 2, comment 1). By fitting the intensity histogram to a Gaussian and taking into account the diffraction limit, we can qualify the heterogeneity between the different flakes.

Focusing on the initial 2H phase for the 2L flakes shown in Fig. 2 at 730 and 455 nm, we find a significantly narrower distribution with a standard deviation of 2.6% for 730 nm compared to a standard deviation of 7.4% for 455 nm. This difference is substantially larger than the expected difference based on the diffraction limit (<2). The flake measured at 455 nm was therefore more spatially heterogeneous than at 730 nm, which must originate from differences in the preparation of the flakes. We would therefore anticipate faster reaction dynamics under identical conditions.

**Fig. A1. Optical image of MoS₂ using different wavelength.
(from Fig. R9. in Part 2 – section 2)**

Critically, however, all experiments presented in the manuscript using 455 nm (and the added 530 and 600 nm experiments) were done on different flakes (Fig. A1), revealing a trend in the acceleration of the phase transition which persists despite this specific example being chosen for Fig. 2. Assuming that all heterogeneity originates from the exfoliation and transfer onto the glass substrate, the emergence of such a trend is sufficient proof of the existence of the photo redox-driven effect, but absolute numbers on phase front speed etc. will be subject to error bars (see different speeds for 455 nm illumination at 110 mW cm⁻² in the Fig A2 below originating from different regions on the flake).

Fig. A2. Speed of phase engineering with respect to the wavelength of light, evaluated using wavefront speed of bilayer. (from Fig. R8. in Part 2 – section 2)

We therefore conclude that, within our preparation method, we can have high confidence in the observed power and wavelength dependence of the discovered photo-redox patterned pathway.

This discussion is now clarified in the manuscript and added to the supporting information. Especially, we added more detail in RICM analysis section in SI (1.4.2), and now provide a new section in supporting information section (SI section 2.1.) that includes the above discussion and motivate explicitly in the main text that further studies targeting this initial observation are required.

Comment 3: The Raman peaks in Fig. 1d of both 2H and 1T phases have demonstrated strong and clear A_{1g} and E_{2g} peaks. However, such representative peaks are usually dampened after the phase transformation after n-BuLi treatment. Does this suggest incomplete phase transition?

In addition, the J-peaks intensity of 455 nm illuminated sample is not as strong as the 730 nm illuminated sample. This should also be further clarified.

We thank the reviewer for this keen observation, which prompted us to conduct more experiments and carefully examine the literature. As the reviewer correctly mentions, both Raman A_{1g} and E_{2g} are dampened at phase transition to 1T. This has been shown in calculation and experimental works²⁻⁴.

Fig. A3. Raman spectra of monolayer MoS₂ with dependent to the n-BuLi treatment time
(from Fig. R 13. in Part 2 – section 5)

We also observe both peaks are dampened as phase transition to 1T, with respect to the n-BuLi treatment duration, indicating a complete 1T phase transition. We believe we caused confusion in Fig. 1D by plotting the spectra scaled to emphasize the red-shift of E_{2g} peak and emergence of J-peaks. For clarity, in Fig. A3, we show the time-evolution unscaled Raman spectra as a function of treatment time, clearly showing the dampening effect the reviewer was referring to.

The emergence of J-peak is clear indicator of the 1T phase, while the peak position and intensity can vary substantially depending on the sample conditions, which include the effect of washing steps, air exposure and measurement parameters. Even within the same chemical treatment and environment, we found from Raman mapping, that the J₂ peak in 1L 1T-MoS₂ on glass substrate varies in intensity by $\pm 20\%$ (95% confidence interval) (in Part 2 – Fig. R 17). The variation in peak position and intensities is also found in literatures (Fig. A4).

Fig. A4. Raman spectra of 1T phase MoS₂
(from Fig. R 16. in Part 2 – section 5)

We have explored all these effects in detail and included the results in the supporting information (see additional Raman mapping experiments and literature reports in Part 2-Section 5 of this Response).

Briefly, from these additional studies we conclude that the emergence of J-peak is the indication of 1T phase, but that the absolute intensity of these features is not a very good metric to compare how well the 1T phase has been formed – which explains the J-peak discrepancy in Raman spectra between red and blue excitation. It is also worth pointing out that even between dark and 730 nm illumination, the 1T spectra differ by a similar amount in the J₂ and J₃ peaks. Due to this complication, we propose that the emergence of any J peak must be accompanied by a red-shift of E_{2g} peak to clearly assign the 1T phase⁵. It is worth

noting that we have monitored our real-time phase transition inside an air-tight cell, but measured Raman in ambient conditions. Furthermore, for safe ex-situ measurements with stable signals, samples were washed out multiple times using Acetone, isopropyl alcohol (IPA) and deionized (DI) water, for removing the residue of n-BuLi, organic and lithium and lithium oxides on the surface. Without proper washing, the thin layer of MoS₂ gets easily covered with thick organics and lithium oxide layer by air exposure. This process could cause the sample to have stronger E_{2g}, A_{1g} peaks and lower J peaks.

In Fig. A5, we further conducted ex-situ experiments to correlate the E_{2g} peak shift to the photoluminescence (PL) intensity decrease. When PL drops significantly, we can observe the red-shift of E_{2g} peak (3.8 cm⁻¹). This shift is as same as we achieve by photo-redox patterning (3.9 cm⁻¹), or literature (2.7 cm⁻¹)⁵, showing the robust nature of this feature in assigning the phase transition. This comparison further supports that the red-shift of E_{2g} peak can be used as the indication of 1T phase.

Fig. A5. Ex-situ Raman analysis for monolayer MoS₂ treated with n-BuLi with different time. A. A-exciton PL (around 660 nm) intensity. **B.** Raman E_{2g} peak position. All sample has been washed using deionized water, Acetone, and IPA. (from Fig. R 15. in Part 2 – section 5)

We provide detailed analysis in Part 2 – section 5 with washing effect, Raman mapping data of photo-redox engineered 1T MoS₂ film, and E_{2g} peak shift compared to PL decreases. From the detailed ex-situ Raman analysis, we conclude that the dampened Raman E_{2g}, A_{1g} mode and the emergence of J peak in Fig. 1D clearly shows we successfully change the phase of

MoS₂ to 1T during our in-situ observations, and that any variations in the J peaks are secondary in nature and indicate slight structural differences.

We included unit scale (a.u.) in Fig. 1D and clarified the normalisation in the figure caption.

All information discussed in here and in Part 2 – section 5 have been added to supporting information section (SI section 2.3.)

Comment 4 :The authors claimed that there exists another pathway which proceeds via the surface penetration of Li ions in the center of the flake, according to Fig. 2e. However, how was such penetration formed, and is it random with more experimental evidence?

We thank the reviewer for pointing out this inconsistency in our explanation. Previous literature reported that the lithium cations can intercalate through the intrinsic defects into the few-layer MoS₂, proven by both experiment and calculation⁶. In their study, the authors experimentally confirmed that a sealed-edge MoS₂ flake was uniformly intercalated through the top surface. This was further supported by calculations. In a perfect MoS₂ monolayer, the energy barriers for top-surface lithium intercalation is calculated by 4.03 eV. The energy barriers with existence of single, or double sulfur (S) vacancy remains similar. In contrast, the energy barriers were significantly reduced to 1.30 eV for MoS₂ monolayer with single molybdenum (Mo) vacancy.

The concentration of single molybdenum (Mo) vacancy has been reported in level of 10¹¹ ~ 10¹² cm⁻² ^{6,7}. This is lower than the density of predominant S defects (~10¹³ cm⁻²), but distributed on surface. Therefore, before the ‘activation’ of Li penetration into Mo vacancy happens, the major intercalation path proceeds via the interlayer intercalation. However, when illuminating the flake with above-gap light, it is possible that the surface penetration reaction can be accelerated and become comparable to the intercalation reaction.

Turning to our experimental data, we find that the core of the flake darkens before a ‘sharp’ phase front reaches the core of the flake. However, since the phase front becomes more diffuse at higher powers densities for C-exciton excitation, we cannot clearly assign this observation to a surface penetration, rather than just a surface effect with our current data. In future, larger bilayer flakes could be measured to delineate the effect of surface penetration vs edge-intercalation, but this is beyond the scope of this work.

We have now removed this section of the manuscript as it does not change the overall conclusion of the revised manuscript, which is primarily concerned with ways to accelerate the phase transition using redox-potential matching and light-activation.

Comment 5 :The authors mentioned that the phase conversion of WS₂ is irreversible. Then how was the phase transition of MoS₂? Is it reversible? Based on the discussion of the mechanism, it seems that such 2H-1T transition can be reversed. Experimental evidence is preferred to confirm if such reversion can be achieved.

The phase transition of MoS₂ is indeed reversible and we have now added additional experimental validation as requested by the reviewer. The 1T MoS₂ phase can be converted back to 2H phase by both thermal annealing and laser irradiation^{8,9}. As shown in Fig. A6, the photo-redox patterned monolayer is reversible and converts back to 2H phase during thermal annealing (in Ar/H₂ atmosphere, 300 °C, 3 hours). This confirmed by PL restoration (B), and Raman spectra (D, top). Here, the E_{2g} peak gets narrower and blue-shifted as the phase restores back to 2H by thermal annealing.

The reversibility was also confirmed in a thick exfoliated sample (over 10 layer) using strong laser irradiation. We irradiated a thick 1T sample using a confocal Raman setup (~10 min, 532 nm, 500 mW, focused on diffraction-limited spot (NA = 0.85)). The Raman spectra from the thick 1T MoS₂ return to their 2H phase, as the bottom graph of D (pink line). This shows that the 1T phase MoS₂ made by either conventional process (48h n-BuLi treatment) and photo-redox engineering process is reversible.

Fig. A6. Ex-situ Raman and PL analysis for monolayer MoS₂. **A.** PL of as exfoliated and phase patterned 1T. **B.** PL of photo-redox phase patterned and annealed 1T. **C.** Raman spectra of as exfoliated (bottom) and photo-redox phase patterned 1T (top). **D.** Raman spectra of annealed 1T (photo-redox phase patterned and annealed sample) (top), and laser-irradiated thick 1T sample (bottom). (from Fig. R 12. in Part 2 – section 4)

The reversibility also can be found in the Fig. A4, that shows thermal annealing restores the PL signal and shifts E_{2g} back to original 2H-phase position. Detailed explanation has been added to Part 2 – Section 4.

All information discussed here and in Part 2 – section 4 has been added to supporting information section (SI section 2.15.)

Comment 6: What is the contact property between 1T-MoS₂ and Au electrode? The enhanced photocurrent cannot unambiguously confirm the lower contact resistance by forming the semi-metallic 1T phase. The Schottky barrier height can also affect the carrier transport which can be influenced due to different work function of the contact stack system.

We thank the reviewer for this comment, which enabled us to further strengthen the photo-detector results. For extracting the contact property between 1T MoS₂ and the Au electrode, we further analysed our data using Fowler–Nordheim (FN) theory. FN theory suggests emission behavior of electrons from metallic emitters under an electrical field follows the equation^{10,11} :

$$J_{FNT} \propto V^2 \exp \left[\frac{-8\pi \sqrt{2m} * \varphi_B^{\frac{3}{2}} d}{3hqV} \right]$$

(where V : Bias voltage, φ_B : tunneling barrier height, m : effective electron mass, h : Planck constant, q : elementary charge)

By plotting the J/V^2 against $1/V$, we can obtain the FN plot which reports on two distinct regimes of field emissions¹²:

- (1) Thermionic emission (TE): Direct tunneling dominates (positive slope, at large $1/V$ region)
- (2) FN tunneling dominates: Schottky barrier exists (negative slope, at small $1/V$ region)

Fig. A7. Fowler-Nordheim (FN) plot for monolayer MoS₂ photodetector. A. FN plot for photo-redox phase patterned (PR-PP), and As-exfoliated (2H) MoS₂ photodetector measured in dark. **B.** FN plot for PR-PP, and As-exfoliated (2H) MoS₂ photodetector during 405 nm illumination. (from Fig. R1. in Part 2 – section 1)

Fig. A7 shows the FN plot for our photo-redox phase engineered sample (PR-PP), and no-patterned, as-exfoliated (2H) 1L-MoS₂ photodetector under dark conditions. Here, the device made of 2H 1L-MoS₂ (as-exfoliated, black curve) exhibits both direct tunneling (positive slope) and FN tunneling behavior (negative slope), while the PR-PE device showed only direct tunneling behavior (blue curve). This behavior remained when the photodetector was operated under 405 nm illuminations as shown in Fig. A7 B.

From here, we can conclude that the transport behavior in the PR-PP device switched from FN tunneling (i.e. a Schottky barrier exists) to direct tunneling at this voltage range by applying 2H to 1T phase engineering as illustrated in Fig. A8.

Fig. A8. Schematics for electron tunnelling behavior for phase engineered MoS₂ device, from Fowler-Nordheim (FN) analysis. (E_c : conduction band energy level, E_f : fermi energy level, from Fig. R2. in Part 2 – section 1)

The work function of 2H MoS₂ is around 4.3 to 4.5 eV depending on its surrounding environment^{13,14}, and 1T MoS₂ varies from 4.4 eV to 5.8 eV depending on the adsorption of hydrogen functional groups¹⁴. In literature, it has been reported that phase engineering switches Schottky behavior to Ohmic-like behavior by renormalization of the band alignment which dramatically enhances the transport properties^{15,16}. Given the enhanced transistor performance in same device geometry^{15,16}, the enhanced photoresponsivity in our study can also be attributed to Ohmic behavior.

We added Fig. A8 in Fig. 3B in manuscript, and all information here and Part 2 – Section 1 in this letter to supporting information section (SI section 2.10.)

Comment 7: One last comment concerning the practical application, the demonstrated approach utilizes optical illumination with different wavelength and power intensity, which can also introduce significant impact on the intrinsic property of the TMD materials as well as the TMD-based electronic devices. The authors are suggested to make a more comprehensive discussion regarding this issue.

Based on the expanded analysis of our photo-detector and the verification of Ohmic-like contact properties, we are confident that the presented photo-redox phase engineering

strategy is applicable, at least up to a power density of 1 W cm^{-2} without any detrimental effects caused by the illumination. We monitored the phase transformation at this power density and found it to occur in under 10 s for monolayer MoS_2 , rapid enough to be of relevance in practical applications. However, we note that actual device structures are unlikely to be fabricated on glass substrates. Further device-focussed studies are needed to explore the substrate effect and any intrinsic changes caused by illumination in this situation.

Moreover, in our revised manuscript, we have developed an even faster approach to achieve phase transformation, by combining illumination with a polyaromatic organo-lithium system (Part 2 – Section 6). The accelerated phase transformation, accomplished through redox-potential matching dramatically reduces the necessary power densities that are needed to complete the reaction. This new system can be used to rapidly achieve phase engineering in TMDs, with even less light-induced property change in the material.

To put this power density in context, a power density of 1 W cm^{-2} is low enough to not significantly affect the intrinsic material properties of TMD, as highlighted in literature, where the PL of monolayer MoS_2 remained unchanged for up to 50 min when continuously irradiated at 532 nm laser with a power density of $\sim 1769.3 \text{ W cm}^{-2}$, i.e. 1000-fold higher than the powers we use. According to the same study, bilayer MoS_2 is not affected until a power density of $\sim 2653.9 \text{ W cm}^{-2}$ for up to 305 min¹⁷. Therefore, we can conclude that laser density of 1 W cm^{-2} has negligible effect on the material. Furthermore, from our new wavelength dependent measurements, we find that 532 nm exhibits the same mechanism as 455 nm, i.e. represents C-exciton excitation (see section 1 and 2 in part 2).

The power density used in our work is also significantly lower than the damage threshold, which is exploited in layer-by-layer etching, or trimming of the MoS_2 . This occurs at power densities of approximately 3981¹⁷, 625000¹⁸, 21000¹⁹ for MoS_2 , and 26875 for MoTe_2 ²⁰ (unit : W cm^{-2}).

Therefore, as we are at this very low-power range of less than 1 W cm^{-2} , we are confident that our method as does not modify the intrinsic properties of the flake and chemical system.

We have added this discussion to the manuscript and supporting information (section 2.4.).

Referee #2

In this work, the authors report the use of an in situ optical reflectance microscopy technique to monitor the phase conversion of transition metal dichalcogenides from the 2H polytype to the 1T/1T' polytype via reaction with n-BuLi. They find that 455 nm illumination accelerates the polytype conversion considerably, and also enables photo-patterning of polytype domains. This is attributed to a photo-excitation mechanism, in which 455 nm illumination excites the C-exciton, which lowers the electron donation barrier and hence accelerates the polytype conversion. The study is quite detailed and provides a useful picture of the dynamics of polytype conversion under "normal"/dark and photo-accelerated conditions. The comparison of different TMDs increases its usefulness to the 2D materials community and reveals some intelligible trends. However, in my opinion, while technically strong, it seems the general impact of this manuscript at present is a bit lacking for a work to be published. Specifically, the authors do not yet demonstrate very distinctive benefits or scientific advances that result from the marked acceleration of the polytype conversion on its own or the photo-patterning concept (the photodetector demonstration is interesting, but it is not clear if the device performance is a major advance or if the approach improves fabrication/processing [speed, cost, etc] compared to other approaches). In addition, the claims of improvements in safety, scalability, and yield are somewhat difficult to understand, as n-BuLi must still be used.

We thank the reviewer for their endorsement of the strong technical merit of our work and the wide interest of the study in general. We have taken each suggestion the reviewer provided seriously and significantly improved the manuscript in three key aspects:

- Based on our insights into the mechanism, we developed a new chemical system which no longer requires n-BuLi. Instead, we can use safer and inherently faster poly-aromatic systems through redox potential tuning. This discovery removes another major hurdle toward scale up of phase engineering and importantly still benefits from further acceleration due to photo-excitation, enabling laser patterning approaches (part 2- section 6, Section 2.16-17. in supplementary information)
- We have deepened our understanding of the photodetector and show a transition from Schottky to Ohmic-like contact properties as a results of phase patterning with light (part 2- section 1, Section 2.10. in supplementary information). This result, is consistent with performance improvement in previous work which used selective exposure of the contact area with n-BuLi for 48 hours. Our new analysis demonstrated the high quality of contacts formed via photo-redox phase engineering. (section 1.7. in supplementary information for comparing fabrication procedure).
- We have conducted additional experiments to explore the wavelength dependence of the photo-driven process. All above-gap excitation accelerated the reaction, but we uncovered that C-exciton excitation leads to the fastest reaction times due to the unique band structure and a corresponding low exciton binding energy (part 2 - section. 2, Section 2.5. in supplementary information).

General question: How do the properties of the 1T polytype produced under 455 nm illumination and under "dark" conditions differ? The authors state that the conversion under 455 nm illumination results in fewer/no wrinkles. Since the greatest interest in polytype engineering stems from the promise of these materials in device applications, it would be useful to know how the transport properties of 1T-MoS₂ or 1T-MoSe₂ produced via these two different pathways compare, in order to evaluate the impact of the technique presented in this manuscript.

We thank the reviewer for their interest in our work and the insightful general question. The 1T phase produced under 455 nm illumination and under "dark" conditions can be regarded as identical from the point of view of optical properties and electrical contact properties:

1. The Raman spectra (red-shift of E_{2g} peaks, emergence of J-peaks) and decreased photoluminescence intensity reveals that the 1T phase made with two different illumination methods shows the same signal (Fig. 1 C, D, Part 2-Section 5 in this letter).
2. We show that the photo-redox engineered flake (455 nm illumination) can be re-converted to its 2H phase. Here, the PL signal restores back to 2H after photo-redox patterning 1L-MoS₂ by annealing at 300 °C. The same result was achieved under "dark" condition previously⁸ (Fig. B1, detail in Part 2-Section 4, 5 in this letter).

Fig. B1. Ex-situ Raman and PL analysis for monolayer MoS₂. **A.** PL of as exfoliated and phase patterned 1T. **B.** PL of photo-redox phase patterned and annealed 1T (from Fig. R 12. in Chapter 2 – section 4)

3. We further analysed our photodetector data using Fowler–Nordheim (F-N) tunneling model and figured out the transition from a Schottky to an Ohmic behavior in the potential profile when photo-redox phase engineering (455 nm illumination). The same Ohmic behavior has been reported in 48h n-BuLi treatment (dark)^{15,16} (Part 2 – Section 1).

In conclusion, the data on electronic transport properties, optical properties and reversibility shows that the photo-redox engineered 1T phase MoS₂ is the same as 'dark (48-h n-BuLi treatment)' sample. We have added all the detailed information to the manuscript and supporting information. We do note, that the photo-redox based samples do not show the same 'wrinkling' as the samples prepared under 730nm illumination. At this time, we are not able to comment on the changes in mechanical properties that might be caused by the meso-scale wrinkling effects, but this could be explored in future work focused on the large area mechanical properties of these materials.

Comment 1 : Figure 2, discussion of histogram analysis: Why are the intensity distributions after 1T conversion considerably sharper under 730 nm illumination compared to 455 nm illumination? Although only one intensity peak is observed under 455 nm illumination, could

the broadness of this intensity suggest some inhomogeneity that the authors have not discussed?

The reviewer raises an important point that we have now examined more closely, and which enabled us to be more quantitative in the description of heterogeneity.

The histogram maps presented in Fig. 2 are computed by masking the relevant section of the flake and generating intensity histograms for each masked recorded image. The mean intensity of the first image is then computed and used to normalise all subsequent intensity histograms to centre them around 1.

As the reviewer correctly pointed out, the width in this histogram representation is a measure of heterogeneity in the recorded pixel intensities. However, there are several other factors that need to be considered before a statement about heterogeneity during a phase transition can be made:

1. The intensity histogram of a portion of a camera-recorded image is in the first instance a convolution of the actual intensity value and the noise characteristics of the camera. We verified shot-noise limited performance and can conclude that the noise broadening is well below 1%, causing no significant broadening to the normalised histograms.
2. The diffraction limit in an optical microscope acts as spatial broadening, effectively cutting off high-frequency spatial features in an image. The minimum resolvable feature size, d , can be calculated from the Abbe formula in our experimental setup to be: $d(730 \text{ nm}) = 260 \text{ nm}$ and $d(455 \text{ nm}) = 162 \text{ nm}$. Therefore 455 nm imaging resolves more high-frequency components and is more sensitive to the morphology of the flake and therefore the preparation procedure. Given that the flakes in our experiments are mechanically exfoliated and not perfectly smooth, we expect generally a broader intensity histogram at 455 nm compared to 730 nm.
3. The normalisation procedure ensures that all images are internally referenced to the optical response of the 2H phase at a given wavelength. However, both the 2H and 1T phases are sufficiently different that they exhibit different optical responses at different wavelengths. This manifests primarily in a different degree of intensity loss during the phase transition, which we observed in Fig. 2. Due to the normalisation procedure, we would expect that the histogram narrows by the same fraction as the optical intensity of the flake reduces during the reaction.

To explore if we can make any claims on heterogeneity, we have now analysed the width of the normalised histograms, as shown in the Fig. B2.

Fig. B2. Histogram fit analysis of bilayer data represented in Fig. 2. **A,B** standard deviation retrieved from a Gaussian fit to the histogram maps shown in Fig. 2 for 730 and 455 nm,

respectively. For 730 nm (**A**), the 2H distribution is fitted to a single Gaussian function until 400 min, after which the 1T bi-peaked phase was visually established and fit to a sum of two Gaussian functions. For 450 nm (**B**), we only fitted the beginning (2H) and end (1T) state to a single Gaussian function, since the fast peak broadening and shifting during the transition (grey box) cannot be accurately described with a single Gaussian.

Comparing the distribution of the 2H phase at 730 and 455 nm, we find a significantly narrower distribution for 730 nm (A) compared to 455 nm (B). This difference is larger than the expected difference based on the diffraction limit. The flake measured at 455 nm was thus more spatially heterogeneous than at 730 nm, likely due to subtle differences in the exfoliation/preparation. This can also be visually identified by comparing the flake images, which suggest a ‘clean’ single-domain flake measured at 730 nm, compared to a multi-domain flake at 455 nm.

Critically, however, all experiments presented in the manuscript using 455 nm (and the added 530 and 600 nm experiments) were done on different flakes, revealing a trend in the acceleration of the phase transition which persists despite this specific example being chosen for the Fig. 2. The Fig. B3 shows that the phase front velocity accelerates by >2 orders at 455 nm. This is significantly larger than the variability between different flakes, which can be estimated from three different regions of the flake presented in Fig. 2 (455 nm, 110 mW cm⁻²), showing ~40% variability, or different bilayers measured at 600 nm, differing by ~30-50%.

Fig. B3. Speed of phase engineering with respect to the wavelength of light, evaluated using wavefront speed of bilayer. (from Fig. R8. in Part 2 – section 2)

We can therefore conclude that, within our preparation method, we are sufficiently consistent to not wash out the observed power, and wavelength dependence of the discovered photo-redox patterned pathway.

Turning now to the 1T phase, we find at 730 nm (A) a subtle increase in width from 2.6 to ~4%, counteracting the expected trend, which should see the standard deviation reduce to ~1.6% (see point three). This step change is significant and we can therefore conclude that at 730 nm (and also in the dark), the phase transition furnishes undesirable additional

heterogeneity. We point out, however, that this may well be substrate dependent: Since actual devices using these flakes are not normally made on glass, which has a relatively weak adhesion, strain effect can most likely be more easily triggered, causing wrinkles and distortions to occur more readily. Future studies on different substrates are motivated to explore this effect in more detail.

The 1T phase in 455 nm (B) shows a reduction in width from 7.4% in the 2H phase to 4.8%. Since the 1T phase exhibits ~40% of the 2H intensity (see Fig. 2), we would expect a width reduction to ~3% if the heterogeneity of the flake would not change during the reaction. While we reduce the standard deviation, we do not achieve this boundary, indicating that the reaction is under 455 nm illumination causes additional heterogeneity across the flake. Critically, however, this trend is significantly better (less heterogeneous) than the behaviour observed at 730 nm, especially considering that we started of with a more heterogeneous flake at 455 nm.

Taken together, this histogram analysis enables us to make several additional, and valuable points, and we thank the reviewer for suggesting this analysis.

We added more detail in RCM analysis section in SI (1.4.2), and now provide a new section in supporting information section (SI section 2.1., 2.5) that includes the above discussion and motivate explicitly in the main text that further studies targeting this initial observation are required.

Comment 2 : Beginning on page 5: The discussion of the intensity values would be easier to follow if the authors specified more clearly in the main text that all of these values refer to normalized intensity, and what the normalization factor is.

We apologise for this confusion. We have now clarified this aspect throughout the manuscript and in the method section, with direct reference to the extra section added in response to comment 1.

Comment 3 : Page 6, 2nd and 3rd full paragraphs: The authors state that "Li-ion intercalation through the surface layer of the 2L" is "not dependent on later Li-ion diffusion through vdW gap." What is the mechanism of this surface intercalation - do the authors think it occurs through defect sites? In addition, there appears to be a logical discrepancy with the statement at the bottom of the page about the power dependence study, where the authors state that the higher fluences demonstrate "that the process is governed primarily by a diffusion limited reaction profile." What is the nature of the diffusion process the authors refer to here?

We thank the reviewer for pointing out this inconsistency in our explanation. Previous literature reported that the lithium cations can intercalate through the intrinsic defects into the few-layer MoS₂, proven by both experiment and calculation⁶. In their study, the authors experimentally confirmed that a sealed-edge MoS₂ flake was uniformly intercalated through the top surface. This was further supported by calculations. In the perfect MoS₂ monolayer, the energy barriers for top-surface lithium intercalation is calculated by 4.03 eV. The energy barriers with

existence of single, or double sulfur (S) vacancy remains the similar. In contrast, the energy barriers were significantly reduced to 1.30 eV when MoS₂ monolayer with single molybdenum (Mo) vacancy.

The concentration of single molybdenum (Mo) vacancy has been reported in level of $10^{11} \sim 10^{12} \text{ cm}^{-2}$ ^{6,7}. This is lower than the density of predominant S defects (in 10^{13} cm^{-2}), but distributed on surface. Therefore, before the 'activation' of Li penetration into Mo vacancy happens, the major intercalation path proceeds via the interlayer intercalation. However, when illuminating the flake with above-gap light, it is possible that the surface penetration reaction can be accelerated and become comparable to the intercalation reaction.

Turning to our experimental data, we find that the core of the flake darkens before a 'sharp' phase front reaches the core of the flake. However, since the phase front becomes more diffuse at higher powers densities for above-band-gap excitation, we cannot clearly assign this observation to a surface penetration, rather than just a surface effect with our current data. In future, larger bilayer flakes could be measured to delineate the effect of surface penetration vs edge-intercalation, but this is beyond the scope of this work.

We have now removed this section of the manuscript as it does not change the overall conclusion of the revised manuscript, which is primarily concerned with ways to accelerate the phase transition using redox-potential matching and light-activation.

Regarding the distinction between charge transfer and diffusion limited regimes, we have now clarified this statement in the text to highlight that we base this assignment on the sharpness of the phase boundary, which, at higher powers, gets markedly more diffuse, indicating a deviation from a strict charge-transfer process.

Comment 4 : Page 7, 1st paragraph: The statement "by accelerating the reaction, chemical safety risks associated with the pyrophoric nature of n-BuLi can be mitigated" seems rather misleading and unsubstantiated. In the "photo-redox" process proposed, n-BuLi must still be used. The risks associated with n-BuLi originate more from its handling than from the reaction time in question. If the amount or concentration of n-BuLi could be reduced through illumination, that might support such a claim. As is, the authors should modify this statement.

We thank the reviewer for their suggestion to explore other, safer, chemicals. In revised manuscript, we added new experiments which accomplish phase engineering of TMDs without the need of pyrophoric n-BuLi. Instead, we exploit a new polyaromatic chemical which is safe, fast, easy to use, and even further accelerates the phase transformation.

In Fig. 4, we have provided comprehensive insights into phase engineering by establishing a correlation with in-situ optical data and electrochemical potential. We also emphasized the significant role of chemical potential in this process. For MoS₂, the electrochemical potential associated with the phase transition lies around 1.1 V (versus Li/Li⁺), and another lower potential appears around 0.57 V (versus Li/Li⁺), assigned to conversion or amorphization via irreversible decomposition. Consequently, the redox potential of the chemical utilized for the phase transition must be located within these two potentials. Based on this understanding and a desire to eliminate pyrophoric n-BuLi (redox potential of 1V), we synthesized organolithiation

agents to phase engineer TMD materials. We prepared a new reagent by reacting Anthracene or Pyrene with lithium metal in tetrahydrofuran solvent, furnishing a solution of aromatic sigma-radical anions counterbalanced by Li cations.

This new reagent enabled efficient and rapid phase engineering to 1T even for thick flakes in < 30 min (see Fig. B4, A). Moreover, combination with the discovered photo-redox process (445 nm illumination, 130 mW/cm²) with the newly synthesized anthracene-Li system further accelerates the reaction to completion in ~10 s for thick flakes (see Fig. B4, B). This is significantly faster than with n-BuLi, even under illumination. The redox-based organolithiation agent (PAH-Li) is easy to use, safe (non-pyrophoric), and can perform rapid phase engineering to MoS₂ and MoSe₂. We anticipate that it will replace n-BuLi and could make whole process faster, safer and greener and aid in the scale up of phase engineering.

Fig. B4. Photo-redox phase transition using Anthracene-Li system. **A.** Ex-situ optical analysis for thick ($\gg 10$ layers) MoS₂ flake treated with the anthracene-Li for 600 s without illumination. The dashed line shows clear wave front. **B.** Ex-situ optical analysis for thick MoS₂ flake treated with the anthracene-Li for 10 s with 445 nm illumination. **C.** Raman spectra measured from Sample in **A**. Each spot is designated on **A**. **D.** Raman spectra measured from Sample in **B**. Each spot is designated on **B**. (scale bar for all figure = 5 μ m) (from Fig. R 19. in Chapter 2 – section 6)

Method of anthracene-Li: Anthracene was dissolved in anhydrous tetrahydrofuran (THF) solvent to its maximum solubility. Stoichiometric lithium metal (lithium granular) was added to the anthracene-THF solution. We exfoliated a thick MoS₂ ($\gg 10$ layers) on glass substrate and placed it in an empty glass vial under ambient conditions before filling it with the anthracene-Li solution. One vial (with sample image in the bottom of Fig C) was allowed to react with the solution for 10 min without any light, while another vial (with sample image in the bottom of Fig D) was exposed to 445 nm LED illumination (SOLIS-1D, power density = 130 mW cm²) for 10 seconds. We washed both samples and monitored the reaction using optical images of

each thick flakes *ex-situ*. In Fig A and B, we observe clear wavefronts as we observed with n-BuLi treated thick-MoS₂²¹. From the wavefront shape, we can roughly calculated a phase front speed for comparison. The wave front speed of flake A (dark) showed ~ 5 nm s⁻¹, while flake B (illuminated) showed a markedly enhanced speed of ~ 5 μm s⁻¹. These front speeds are significantly faster than what we could achieve using n-BuLi with illumination and enable the process even in thick flakes, where the combination of redox matching and above-gap illumination accelerates the reaction significantly.

This detailed results have been added to the Fig. 5 in main manuscript and explanation here and the part2 – section 6 added in SI (section 2.16. 2.17.).

Comment 5 : Page 9, The authors state the utility of the process described in patterning "few-layer" TMDs. The main examples in the paper are for 2L and 1L samples. One might expect that the lower surface area-to-volume ratio for increasing layer numbers would hamper the surface-mediated mechanism proposed for polytype conversion under above-bandgap illumination. In addition, the evolution of the bandgap from direct to indirect going from monolayer to multilayer flakes is not discussed. What, for example, is the effect of illumination and polytype conversion in 2L samples that are not connected to a 1L region? It would be helpful if the authors could comment on the utility of this approach for different thicknesses of MoS₂ or MoSe₂ flakes. (Indirect gap : 830nm, but similar absorption graph)

The demonstration of mono- and bi-layer examples was motivated by their use in electronics, where phase engineering has its greatest potential application. But the reviewer is absolutely right that the study of the light-mediated process we discovered for different layer thicknesses and exciton type will widen the generality of the findings. We have therefore conducted several new experiments, including on thick flakes, to show the generality of our approach, which are in part 2-section 3 in this reply letter, and now included in the main manuscript and in the supporting information (section 2.6.) We hope these additional experiments help clarify the effect of the proposed mechanism. We also clarify the edge/terrace dependence under illumination.

Exciton dependence: In response to all reviewers we have carried out experiments at two additional wavelengths at 530 nm (nominally off-resonant) and 600 nm (resonant). These results are summarised in Section 2 in Part 2 below, which are included in the revised supporting information.

Briefly, 600 nm excites the B-exciton, which shows a high exciton binding energy and has a direct bandgap for the monolayer in 2H-MoS₂. We find that optical excitation of this exciton during phase transition in n-BuLi leads to an accelerated phase transition compared to below-band gap illumination and dark conditions. However, the acceleration is significantly lower than for 455 nm illumination, which excites the C-exciton of MoS₂. This trend applies for both mono and bilayers. Notably, throughout the experiment, the phase front remained sharp indicating charge transfer limitations

At 530 nm, the phase transformation was accelerated compared to 600 nm and proceeded with more diffuse phase fronts, similar to 455 nm low power illumination, suggesting a deviation from a strict charge-transfer process. 530 nm excitation thus behaves like C-exciton excitation, albeit still slower.

Taken together these results demonstrate that the critical part toward accelerating the phase transition is the availability of the material to create free charges under photo-excitation. Here, the C-exciton outperforms the B-exciton (and also the A-exciton given their close electronic nature), which we attribute to band nesting effects.

Edge dependence: Fig. 2C reveals a pronounced difference in the phase conversion onset (or phase nucleation) dependent on whether the intercalation originates from a glass/bilayer edge, or a monolayer/bilayer terrace. While not mentioned explicitly in the text, Fig. 2D provides a comparable scenario whereby the top left part of the flake is an edge (glass/2L), while the bottom right a Terrace (1L/2L). From the images in Fig B5, we find that the triangular edges of the bilayer flake and strong defect sites in the center act as primary nucleation sites, as shown in red circle:

Fig. B5. Visualisation of several 1T nucleation sites (red circles) in 2L under 455 nm illumination. (from Fig. R 10. in Chapter 2 – section 3)

Notably, all nucleation sites indicated above occur roughly at the same point in time with no significant delay irrespective of whether the structure is a terrace or an edge. Further, once nucleated, the phase transformation proceeds through that phase, pushing a (more diffuse) phase boundary along until the entire flake is converted. This behaviour is markedly different than the flake examined under 730 nm illumination, where the 1L terrace nucleated first along the terrace, rather than in its corner.

These results suggest that photoexcitation provides sufficient energy to overcome the difference in nucleation activation energy barriers between edges and terraces and that it prefers nucleation along exposed ('pointy') features of the flake.

Thickness dependence: To explore how the observed photo-redox process occurs for thicker samples of more than 2L, we conducted experiment on thicker flakes:

- At 600 nm, we explored a terraced structure with increasingly thicker layers

Fig. B6. Snapshot of the phase transition in terraced-structure MoS₂ visualized at 600 nm. (Scale bar = 5 μm) (from Fig. R5. in Chapter 2 – section 2)

Noticeably, we observed nucleation of the 2L with a single wavefront moving from the bottom side of the image across the whole layer, independent to the numbers of layers. This suggests a layer-by-layer approach between successively thicker layers. Importantly, this highlights that even thick layer structures can be converted more rapidly with light exposure in such structure, as the mechanism remains the same, that is, the activation barrier toward nucleation is lowered across the surfaces, helping to move the phase front along.

Two caveats remain: Firstly for the thicker layers in this experiment we observe more wrinkles and in some other cases even delamination, most likely due to internal strains imposed by the surrounding environment. Secondly a bulk sample with no terraces will not benefit from the photo-redox effect as much as a terraced structure.

- At 530 nm, we carried out measurements on a terraced structure with a large 3-4L flake as shown below:

Fig. B7 Snapshot of the phase transition in MoS₂ visualized at 530 nm. (Scale bar = 5 μm). (from Fig. R 11. in Chapter 2 – section 3)

As before, the 1T phase nucleates at corner of the 4L flake (right hand side in the Fig. B7 and move through this nucleation point across the flake. Interestingly, after about 84 min, we observe distinct intensity regions deep inside the flake, labelled ‘a’ (orange) and ‘b’ (purple), which connect the 2H and converted 1T phase. These intermediate region likely correspond layered structures in which 1 or 2 of the 4L structure exhibit 1T character. We highlight that for a large 2L structure illuminated at 600 nm no such phase terracing was observed.

These results outline that the process of a photo-redox phase transition is functional in thicker flakes but proceeds via a more complex mechanism that depends on the terracing of the structure itself. This effect may be used to vertically inscribe heterogenous 1T/2H structures, but more research needs to be conducted to explore the optimised parameters for this process. Critically, it highlights that the effect of illumination is not just confined to the surface, but proceeds also via a bulk effect.

Generalisation to thicker systems: To enable rapid conversion of thick flakes in simpler way, photo-redox pathways alone are not sufficient. Instead, we show in our manuscript how redox matching can be utilised to ensure such samples can be converted. (previous comment and part 2-section 6, or, section 2.16 in supplementary information)

Referee #3

The manuscript reports on an interesting study of photo-assisted 2H-1T phase transition in TMDCs using n-BuLi chemical lithiation. The work expands the current knowledge on the mentioned phase transition through the introduction of the technologically relevant blue illumination driven process, which can also be used for an efficient and simple patterning. The study is mostly conducted via in-situ optical reflectance experiments in a controlled environment. On the downside, even though the advancement in the field relates almost exclusively to the photo-assisted process, its explanation is vague and relies on an assumption which the authors do not try to prove (e.g. through the use of different illumination wavelengths that would probe other excitons, or, in contrast, be away from their energies). Similarly, the part on the differences between Mo and W-TMDCs could be strengthened by a reversibility study. Several other issues are listed below. To summarize, in spite of an undeniable appeal of the study to a large readership in the field of 2D materials, the limited novelty and depth of the work make the manuscript in the current form not well suited.

We thank the reviewer for their time and the constructive questions and comments. Below, we added more supporting evidence to address each comment.

But before the point by point response, we would like to highlight that we have expanded our study significantly and developed a new chemical systems, which together with optical illumination achieves the 1T phase transformation in several seconds even for thick flakes, outperforming our previous work by ~3 orders of magnitude. We hope and believe that this novel discovery, backed by a more in-depth analysis and more mechanistic work will be of interest to a large readership (part 2 – section 6, Fig. 5 in main manuscript and section 2.16-17 supplementary information).

Comment 1 : The explanation of the photo-assisted process is vague and should be strengthened at least by exploring the effects of other illumination wavelengths in/out resonance with (other) excitons. Ideally, this hypotheses should also be supported by theoretical calculations.

We thank the reviewer for their suggestion and have now further explored the wavelength dependence of the photo-assisted process by measuring the phase transformation speed at 530 nm, which lies nominally between the B and C exciton of MoS₂, and also at 600 nm, resonant with the B exciton. The results further support our mechanistic picture and the full discussion of these results is now clearly expressed in the manuscript and presented in the supporting information and below in Part 2 - section 2, or section 2.5 in supplementary information (also see reviewer 2. Comment 5).

Briefly, for resonant B-exciton illumination at 600 nm, we adjusted the power density to generate approximately the same number of carriers generated at 455 nm (110 mW cm⁻²). We find that the phase transformation at 600 nm displays a clear sharp phase front, which moves ~2-3 fold faster across the flakes as compared to the dark/730 nm measurement. This values is, however, ~10 times slower than the equivalent phase front speed at 455 nm (see Fig. C1 below).

Mechanistically, the existence of a sharp phase boundary implies a charge-transfer limited reaction, but the enhanced front speed suggests a mild acceleration due to photo-excitation. This is consistent with the notion that B-exciton excitation generates tightly bound excitons with a large exciton binding energy (~ 0.4 eV), preventing efficient formation of free charges, needed to accelerate this charge-transfer limited reaction. In turn, C-exciton excitation at 455 nm must generate more free charges, in agreement with literature reports, further accelerating the reaction.

Turning to 530 nm excitation, we set the power density to 510 mW cm^{-2} , equivalent to the higher power measurement at 455 nm presented in Fig. 2. We observed a sharp, albeit slightly more diffuse phase front, with a velocity of 1 nm s^{-1} . This is 2 orders of magnitude slower than the equivalent experiment at 455 nm. However, it compares well in value and behaviour to the 455 nm experiment carried out at a power density of 20 mW cm^{-2} .

Compared to 600 nm excitation (360 mW cm^{-2}), the 530 nm phase front is ~ 3 fold faster, despite only a 1.5 fold increase in power density and a lower overall absorption coefficient. The phase front is also less sharp compared to 600 nm, suggesting mechanistic differences.

Fig. C1. Speed of phase engineering with respect to the wavelength of light, evaluated using wavefront speed of bilayer. (from Fig. R8. in Part 2 – section 2)

Taken together, 530 nm excitation behaves like C-exciton excitation, albeit with slower overall dynamics and a higher power density requirement. This can be, in part, rationalised by decomposing the absorption spectrum of MoS₂ into the individual excitonic resonances (see Fig. C1). Here 530 nm excitation accesses low-energy tail states of the C-exciton (0.38 eV below the absorption maximum), and barely overlaps with the B-exciton. The non-linear

scaling of the phase front speed with the absorption cross section further suggests that a charge generation barrier has to be overcome to generate free charges.

The photo-assisted effect described in our work is therefore universally applicable for above bandgap excitation in TMDs, but the power densities required to reach a desired conversion time are related to the underlying band structure, where C-exciton excitation is significantly faster than B-exciton excitation. We attribute this to the free charge carrier generation rate, which is enhanced for the C-exciton due to band nesting compared to the B-exciton. Within the C-exciton band, lower-energy excitation results in relatively slower phase front speeds not matched by the absorption cross section, suggesting a thermally activated pathway for charge carrier generation²².

Further insights into this mechanism will require theoretical insights capable of accurately capturing the C-exciton electronic structure. While such calculations are beyond the scope of our work, we now explicitly motivate them in the discussion of the manuscript to place the findings of our work on a more atomistic footing.

Comment 2 : The reversibility of the phase transition in Mo-TMDCs could be shown by adjusting the lower cut-off potential in the CV, and, then, the material could be analysed *ex situ*.

We thank the reviewer for this comment. The electrochemical control for degree of lithiation affects both structural and phase properties of MoS₂ and has been studied in previous literature, and we have now included relevant references the background literature in the main manuscript²³.

Briefly, Wang and colleagues studied *ex-situ* characterizations of MoS₂ stopped at different lithiation voltages. In this study, MoS₂ was electrochemically lithiated at different voltages of ¹around 2.1 to 1.8 V (\approx Li_{0.02}MoS₂), ²1.5 V (\approx Li_{0.07}MoS₂), ³1.2 V (\approx Li_{0.28}MoS₂), and ⁴1.1 V (\approx Li_{0.85}MoS₂) (all V versus Li/Li⁺), respectively. Firstly, the layer spacing by electrochemical lithium intercalation was confirmed by transmission electron microscopy (TEM) and the XPS and Raman analysis showed the phase identify of each step as shown in the table C1 in below:

Voltage (V) (versus Li/Li ⁺),	Interlayer spacing (TEM) (Pristine: 6.45 Å)	Phase (XPS Mo3d _{5/2} peak)	Raman
2.1 to 1.8 V (\approx Li _{0.02} MoS ₂)	Similar to pristine, very little expansion (6.50~6.54 Å)	2H (228.7 eV)	Same as pristine MoS ₂
1.5 V (\approx Li _{0.07} MoS ₂)		2H (228.6 eV)	
1.2 V (\approx Li _{0.28} MoS ₂),	Layer expanded (7.25 Å)	Additional peak appeared for 1T (228.2 eV) 1T composition increases at 1.1 V	E _{2g} peak shifts, J peak arises
1.1 V (\approx Li _{0.85} MoS ₂)	Layer expanded (7.21 Å)		

Table C1. *ex-situ* characterization of electrochemically lithiated MoS₂

These results are further supported by in-situ XRD, and in-situ microcell study^{24–26}. From these studies it can be shown that a cut-off potential of around 1.2 V is inducing structural phase transition. This is well matched with our data that the electrochemical potential window of 1.15 V-0.55 V induces the structural phase transition. Therefore, the 1T phase MoS₂ we made in chemical lithiation or photo-redox process is identical to electrochemically synthesized 1T MoS₂ at a Voltage of around 1 V.

MoS₂ in 1T phase is metastable in ambient air and can be reversed back to 2H phase by thermal annealing or laser irradiation. In Part 2 – Section 4, which is now part of the supporting information, we show detailed experimental data on reversibility of our 1T phase MoS₂ for both annealing and laser irradiation strategies (see also response to Reviewer 1, comment 5).

Comment 3 : One very important and confusing issue relates to the phase transition in 1L MoS₂. Page 3 states that the monolayer region is also successfully converted to 1T, but the reference points to Supplementary figures that do not seem to show anything like that (e.g., suppl. Fig. 5 shows a Raman spectrum only of a disordered 1L, with no clear signs of a 1T phase). The statement of the monolayer being converted to 1T is then made several other times, e.g., in relation to the photodetector. However, Figure 2 and its discussion clearly documents that the only process taking place on 1L is the surface charge, not the phase transition, which is shown only for 2L. This ambiguity should be thoroughly sorted out in the whole manuscript.

We apologise for the confusion caused in this description and have now thoroughly revised the manuscript and expanded the supporting information to clarify this notion. We agree, that based on the RICM results alone, it is difficult to ascertain if the phase transformation has taken place for the 1L, or if it is only a surface charge effect. In fact, these two processes might very well happen at the same time and an initial darkening of the 1L phase is observed in Fig. 2. However, due to this confusion, we sought alternative experimental support which we have further clarified in the manuscript and main text:

- The experimental Raman signature of monolayer 1T MoS₂ is dependent on washing procedures, air exposure and measurement parameters. We have systematically explored these steps and suggest that the 1T phase in monolayer MoS₂ what we made inside the *in-situ* cell can be assigned as 1T phase if:
 - the E_{2g} peak red-shifts (~3.8 cm⁻¹) and reduced intensity of A_{1g}, E_{2g} mode compared to the 2H phase at *ex-situ* measurement in ambient using washed-sample.
 - The PL is quenched significantly.
 - J peaks emerge.
- We have verified all three parameters in all experiments confirming successful 1T phase conversion. More details are provided in Part 2 – section 4, and also in answers to Reviewer 1 comment 3.
- We verified that the 1T polytype produced under 455 nm illumination exhibits reversibility back to its 2H phase. Here, the PL signal restores back to 2H at photo-redox patterned monolayer by annealing at 300 °C, as same as what reported with “dark” condition⁸.

Fig. C2. Ex-situ Raman and PL analysis for monolayer MoS₂. **A.** PL of as exfoliated and phase patterned 1T. **B.** PL of photo-redox phase patterned and annealed 1T (from Fig. R 12. in Part 2 – section 4)

- Lastly, further analysis on the photodetector data using Fowler–Nordheim (FN) tunneling model revealed the transition from a Schottky to an Ohmic-like contact behavior photo-redox phase engineering (455 nm illumination). The same Ohmic behavior has been reported in 48h n-BuLi treatment (dark)^{15,16} (Part 2 – Section 1).

In conclusion, based on this data on electronic transport properties, optical properties and reversibility, we are confident that the photo-redox engineered 1T phase in 1L-MoS₂ is the same as ‘dark (48 hour n-BuLi treatment)’ sample, even though the transition from surface charge to 1T in 1L-MoS₂ cannot be resolved clearly with RICM. We have clarified this aspect throughout the manuscript and expanded the supporting information (section 2.3. for Raman analysis, 2.15 for reversibility, 2.11 for FN tunneling analysis).

Comment 4 : Concerning the proposed mechanism of the through-the-layer lithiation for WS₂, the authors should also consider and discuss the possibility that the lithiation directly leads also to an irreversible disorder connected with this process.

We examined previous transmission electron microscopy (TEM) studies to understand this connection. In these studies, WS₂ was directly contacted with lithium or lithium oxide^{27,28}. This mimics the effect of very fast lithiation and drives the reaction to phase conversion (i.e. irreversible decomposition). TEM analysis revealed the coexistence of 2H-WS₂, 1T-Li_xWS₂ and Li₂S, highlighting that an intermediate phase between 2H-WS₂ and Li₂S, namely 1T-WS₂, exists and the decomposition is most likely a two-step process.

Fig. C3. Optical image and Raman spectrum of n-BuLi treated, and Anthracene-Li (AnLi) treated WS₂. **A.** Optical image of n-BuLi treated exfoliated thick WS₂. Shrinking core-type phase front is observed. **B.** Raman spectra of 1T WS₂ (red, from outer ring region in A) and 2H WS₂ (black, from inner region in A) **C.** Optical image of AnLi treated exfoliated thick WS₂. Two wave fronts are observed. **D.** Raman spectra of 1T WS₂ (red, middle ring region in C) and 2H WS₂ (black, from inner region in C), and converged region (blue, out most region in C). (Scale bar for A, C = 20 μm)

To further explore this regime, we turned to our newly discovered chemical system using an anthracene-lithium (AnLi) solution (Part 2- Section 6). While the conversion of WS₂ in n-BuLi is extremely slow and has a very low yield, the use of AnLi provides a larger driving force (~0.9 V), making it possible study both voltage plateaus shown in Fig. 3 more clearly.

When using n-BuLi in the dark, we observed single phase front moving in a shrinking-core fashion inside the flake (Fig. C3 A). Raman confirmed that the outer region changed to the 1T phase (red in Fig C3 B), while the core remained 2H (black in Fig C3 B), with no decomposition products detected.

The use of AnLi driven the reaction further to conversion (decomposition). Optically, we can identify three concentrically located phases which can be assigned by Raman spectroscopy to 2H in the center, a converted phase (outer ring in Fig. C3 C, blue in Fig. C3 D), and an intermediate 1T phase (middle ring region in C and red graph in D). This demonstrates that lithiation of WS₂ follows a two-step process and direct conversion is not the most preferred pathway.

While illumination can alter the activation barriers along the different reaction pathways, it is unlikely to change the fundamental mechanism. Mechanistically, this is further supported by our wavelength-dependent studies on MoS₂, which highlight that the acceleration of the

reaction is a direct consequence of enhanced charge production, effectively resulting in an increased driving force/lower activation barrier, rather than a change in the potential response.

We have added this discussion to the supporting information (section 2.16.)

Comment 5 : Page 5 states “This discrepancy likely originates from different activation barriers along these two interfaces,...”. The authors should expand on why should the activation barriers be different.

We apologize for not explaining this aspect of the work carefully enough. Our experiments show that different sections of a flake can exhibit different nucleation times. For example, the middle region in the Fig. C4 in below (dashed-blue line) shows an earlier reaction than the other regions (marked with star). Given that all three flake segments have been prepared similarly and exhibit the same glass/interfaces, this discrepancy must arise from different activation barriers along these interfaces (reasons may include defect density, strain, substrate interaction variations, etc.). Importantly, while the time for phase nucleation differs, once nucleated, the phase front velocity is the same for all flakes, as also shown in Fig. 2C in the manuscript (identical slope).

Literature calculation studies further propose that the phase transition reaction is more favorable to specific edges, such as the bare Mo-edge or the S-edge with 50% S²⁹. Therefore, we can postulate that different types of edges can cause different activation barrier that determines the time for nucleation of the new phase. This has impact for terraced structures (see response to Reviewer 2, comment 5)

We have now expanded this section to supplementary information (section 2.6.) to reflect this point.

Fig. C4. Snapshot of the phase transition in MoS₂ visualized at 530 nm after 11 min. (from Fig. R9. in Part 2 – section 3)

Comment 6 : One could imagine that a further illumination of the patterned photodetector by blue light (even after the fabrication) might still lead to some mobilization of free Li ions into the photodetector area. Could the authors consider and discuss that possibility?

We apologise for not being clear in how the photodetector measurements were carried out. As in the figure below (Fig. C5), we provide detailed schematics for fabrication and characterization of photodetector. After photo-redox patterning inside the chemical cell, we

took out the sample and washed out adsorbed lithium with deionized water (DI), Acetone, and isopropanol (IPA). This removes all Li from the sample. We then deposited our electrode (Au) using an electron beam evaporation system in high vacuum ($< 10^{-7}$ torr). Therefore, mobilization of free Li ion under photodetector operation will not have an effect on the measurement.

We added the figure below to section 1.7. in supplementary information and clarified the method section accordingly.

Fig. C5. Fabrication and measurement procedure of photo-redox patterned MoS₂ photodetector

Part 2. Newly Included Scientific Data and Explanations

*All information in this chapter has been added to either the manuscript or SI

Section 1. Tunneling behavior analysis between semiconducting (2H)-, and phase engineered metallic (1T) MoS₂ - metal contacts.

In Fig 3 of the manuscript, we show how phase patterning enhances the photoresponsivity of a photodetector. This results from the 2H (semiconductor) to 1T (metal) phase engineering, which has shown its ability to reduce the contact resistance between MoS₂ and metal.

The Schottky barrier for electron injection between MoS₂ and the metal contact is one of the key problems in electronic applications for MoS₂. The Schottky barrier height (Φ_B) between MoS₂ and metal has been reported to the level of few-hundreds meV³⁰⁻³². Lowering the Schottky barrier for reducing the contact resistance, or switching Schottky behavior to Ohmic behavior enhances the transport properties of electronic devices using MoS₂. It has been reported that phase engineering switches Schottky behavior to Ohmic-like behavior by renormalization of the band alignment, which dramatically enhances the transport properties^{15,16}. In our study, the photoresponsivity was enhanced by selectively inscribing a metallic 1T phase at the edge of the semiconducting 2H phase. Given the enhanced metal-semiconductor contact properties in the same device geometry as previously reported^{15,16}, the enhanced photoresponsivity was attributed to the switching the Schottky contact to Ohmic contact behavior by photo-redox phase engineering.

To provide further support, we conducted a detailed analysis of how electrons from the metallic 1T phase is injected to the semiconducting 2H MoS₂ channel. We analyzed the quantum transport behavior of our photodetector data using Fowler–Nordheim (FN) theory. FN theory suggests emission behavior of electrons from metallic emitters under electrical field, following the equation^{10,11} :

$$J_{FNT} \propto V^2 \exp \left[\frac{-8\pi \sqrt{2m} * \varphi_B^{\frac{3}{2}} d}{3hqV} \right]$$

(where V : Bias voltage, φ_B : tunneling barrier height, m : effective electron mass, h : Planck constant, q : elementary charge)

By plotting the J/V^2 versus $1/V$, we can obtain FN plot which suggest two regimes of field emissions¹²:

(1) Thermionic emission (TE) : Direct tunneling dominates (at large $1/V$ region, small bias voltage)

(2) FN tunneling dominates : Schottky barrier exists (at small $1/V$ region, large bias voltage)

Fig. R1. Fowler-Nordheim (FN) plot for monolayer MoS₂ photodetector. A. FN plot for photo-redox phase patterned (PR-PP), and As-exfoliated (2H) MoS₂ photodetector measured in dark. **B.** FN plot for PR-PE, and As-exfoliated (2H) MoS₂ photodetector during 405 nm illumination

Fig. R1 A and B shows FN plot of photo-redox phase patterned (PR-PP), and no-patterned, as-exfoliated (2H) monolayer MoS₂ photodetector under dark (A) and illuminated (B) conditions, respectively. Fig. R1 A shows the FN plot from the device made of 2H 1L-MoS₂ (as-exfoliated, black curve) exhibits both direct tunneling (positive slope) and FN tunneling behavior (negative slope), while the photo-redox phase patterned (PR-PP) device showed only direct tunneling behavior (blue curve). This behavior remained when the photodetector was operated under 405 nm illuminations as shown in Fig. R1 B.

From here, we can conclude that the transport behavior in the PR-PP device switched from FN tunneling (i.e. a Schottky barrier exists) to direct tunneling at this voltage range by applying 2H to 1T phase engineering as illustrated in below.

Fig. R2. Schematics for electron tunnelling behavior for phase engineered MoS₂ device, from Fowler-Nordheim (FN) analysis.

The work function of 2H MoS₂ lies between 4.3 - 4.5 eV depending on its surrounding environment^{13,14}, and 1T MoS₂ varies from 4.4 - 5.8 eV depending on the adsorption of hydrogen functional groups¹⁴. Our analysis using a FN tunneling model demonstrates the

transition from a Schottky to an Ohmic behavior in the potential profile during photo-redox phase engineering. In literature, it has been reported that phase engineering switches Schottky behavior to Ohmic-like behavior by renormalization of the band alignment which dramatically enhances the transport properties^{15,16}. Given the enhanced transistor performance in same device geometry^{15,16}, the enhanced photoresponsivity in our study can also be attributed to Ohmic behavior.

Fig. R1 and related explanations added to supplementary materials (section 2.10.).

Fig. R2 added in Fig. 3B and explanation added accordingly.

Section 2. RICM measurements using Different Wavelength of Light.

In this section, we present RICM results from measurement using different wavelengths at 530 nm and 600 nm. We have chosen different wavelengths following the optical absorption spectra for MoS₂. Ellipsometry was performed on a micro-ellipsometer. The ellipsometry angles were fitted to a thin film model to extract the dielectric constant ϵ of the monolayer MoS₂. The absorption of the monolayer was then calculated following the equation: $Re[-1i \frac{2\pi}{\lambda} d\epsilon]$.

Fig. R3. Optical Absorption spectra for monolayer MoS₂

The absorption spectrum of monolayer MoS₂ shows clear signature of excitonic resonances at ~650 nm (A-exciton), and ~610 nm (B-exciton), and ~450 nm (C-exciton)³³. The C exciton originates from band-nesting effect³⁴, and absorbs more strongly than the B and A excitons. Decomposing the absorption spectrum into individual exciton resonances guided us to select further wavelengths to study. We firstly chose 600 nm to explore potential differences between the C and B excitonic resonance. We note that the A and B excitons are qualitatively similar in nature, and we therefore focused only on the high-energy B-exciton in this study. Secondly, we explore a nominally off-resonant illumination at 530 nm to verify the effect of optical absorbance on this process for correlating the optical absorbance to phase transition speed.

1. RICM measurements at 600 nm.

600 nm excites the B-exciton, which shows a high exciton binding energy and has a direct bandgap for the monolayer in 2H-MoS₂. We find that optical excitation of this exciton during phase transition in n-BuLi leads to an accelerated phase transition (2-3 fold) compared to below-band gap illumination and dark conditions. However, the acceleration is significantly

lower than for 455 nm illumination (>10 fold), which excites the C-exciton of MoS₂. This trend applies for both mono and bilayers. Notably, throughout the experiment, the phase front remained sharp indicating charge transfer limitations

**Fig. R4. Snapshot of the phase transition in bilayer-MoS₂ visualized at 600 nm.
(Scale bar = 5 μm)**

Firstly, we measured a large bi-layer with no monolayer surroundings. The flake was kept in the dark for 596 min, to evaluate the progression of the phase transition under dark conditions. Afterwards, we illuminated the flake with whitelight filtered by a 600 ± 40 nm bandpass filter and a power density of 360 mW/cm^2 . This power density was chosen to excite approximately the same number of excitons as at 455 nm, 110 mW/cm^2 for a direct comparison. As shown in Fig. R4, we observed a clear shrinking core type behavior with a sharp wavefront. This behavior closely resembles the phase transition mechanism observed at 730 nm, suggesting it is charge-limited (730 nm, see Fig. 2A). Here, the edges and cracks across the flake act as 'active' lithiation sides and allow the phase transition to proceed. The velocity of sharp wave front was measured by $\sim 0.24 \text{ nm s}^{-1}$. Comparing to the wave front speed of below-gap illumination (730 nm), which is $\sim 0.08 \text{ nm s}^{-1}$, it is 3 times faster. Illumination at 600 nm thus accelerates the intrinsic reaction mechanism what we measured in 730 nm (charge-limited). We also note the absence of any wrinkles, which is noticeable given the large size of the flake.

Fig. R5. Snapshot of the phase transition in terraced-structure MoS₂ visualized at 600 nm. (Scale bar = 5 μm)

For studying the structure and layer dependence, we conducted second measurement using 600 nm illumination with a terraced-structured MoS₂. Fig. R5 shows the phase transformation of this terraced structure, which shows similar 2L wavefront speed of $\sim 0.32 \text{ nm s}^{-1}$.

Noticeably, we observed nucleation of the 2L with a single wavefront moving from the bottom side of the image across the whole layer, independent to the numbers of layers. This suggests a layer-by-layer approach between successively thicker layers. This highlights that even thick layer structures can be converted more rapidly with light exposure in such structure, as the mechanism remains the same, that is, the activation barrier toward nucleation is lowered across the surfaces, helping to move the phase front along

2. RICM measurements using 530 nm.

The wavelength of 530 nm is above the bandgap, but the optical absorbance is lower than 600 nm. Therefore, we monitor the reaction using slightly higher power than 600 nm (510 mW/cm^2) to compare the reaction in similar optical absorbance.

Fig. R6. Snapshot of the phase transition in MoS₂ visualized at 530 nm. (Scale bar = 5 μ m)

At 530 nm, the phase transformation was accelerated compared to 600 nm and proceeded with more diffuse phase fronts, similar to 455 nm low power illumination, suggesting a deviation from a strict charge-transfer process (Fig. R6).

Fig. R7. Comparison of the phase transition in 2L MoS₂ visualized at 530 nm (top) and 455 nm (bottom).

In Fig. R7, we compare experiments using 530 nm (510 mW cm⁻²) with 455 nm (low power, 20 mW cm⁻²). Clear similarities are revealed both in terms of phase front speed and in terms of the sharpness of the phase boundaries. 530 nm therefore behaves like C-exciton excitation, but slower. This is most likely due to the 0.38 eV lower excitation energy, imposing an excited state barrier on the generation of free charges. More theoretical studies of the exact electronic structure of the C-exciton are required to fully reveal the nature of this effect.

3. Wavelength-dependent photo-redox phase transition speed.

Taken together, these studies reveal a clear picture of the phase transformation mechanism in MoS₂ under optical illumination. The optical illumination using 600 nm (360 mW cm⁻²), 530 nm (510 mW cm⁻²), and 455 nm (20 mW cm⁻²) shows similar wavefront speed as 0.3 - 1.3 nm s⁻¹, compare to dark (0.08 nm s⁻¹) and 455 nm in high power (510 mW cm⁻², 110 nm s⁻¹). However the mechanism of above gap illuminations (530 nm, 455 nm), excitonic resonance (600 nm) and below gap illumination (730 nm) varies slightly. B-exciton excitation behaves

qualitatively similar to 730 nm, with a charge transfer limited reaction mechanism. This behaviour can be attributed to the large exciton binding energy (>0.5 eV), effectively preventing free charge generation. Nonetheless, a mild light effect is revealed suggesting that photoexcitation leads to a small fraction of free charges that can partake in the phase transition. Conversely at 455 nm and 530 nm, the C-exciton resonance dominates the behaviour. In comparison, this strongly supports a higher charge carrier generation yield for the C-exciton compared to the B-exciton, as postulated in previous studies³⁴. Exciting at lower C-exciton energies slows the photo-redox phase patterning reaction down, but not in proportion to the overall absorption cross section, suggesting an excited state barrier to charge generation.

Fig. R8. Speed of phase engineering with respect to the wavelength of light, evaluated using wavefront speed of bilayer.

The phase front speeds retrieved from these studies are reported on Fig. R8 showing that above-bandgap excitation always improves the reaction speed, but that the underlying excitonic structure plays an important role in the absolute acceleration that can be achieved.

Section 3. Image analysis for structural dependent phase transition.

1. Different nucleation point across the flake – Edge dependence

Fig. 2C reveals a pronounced difference in the phase conversion onset (or phase nucleation) dependent on whether the intercalation originates from a glass/bilayer edge, or a monolayer/bilayer terrace. Furthermore, Fig. R9 shows the discrepancy of phase transition in different section of the flake. All flakes show the discrepancy of phase transition in different

section of the flake. The middle region in the Figure in below (dashed-blue line) shows faster reaction then other regions (marked with star). Given that all three flake segments have been prepared similarly and exhibit the same glass/interfaces, this discrepancy must arise from different activation barriers along these interfaces (reasons may include defect density, strain, substrate interaction variations, etc.). Importantly, while the time for phase nucleation differs, once nucleated, the phase front velocity is the same for all flakes, as also shown in Fig. 2C in the manuscript (identical slope).

Fig. R9. Snapshot of the phase transition in MoS₂ visualized at 530 nm after 11 min.

Literature calculation studies further propose that the phase transition reaction is more favorable to specific edges, such as the bare Mo-edge or the S-edge with 50% S²⁹. Therefore, we can postulate that different types of edges can cause different activation barrier that determines the time for nucleation of the new phase.

The images using above-gap illumination (Fig. 2D) provides a comparable scenario whereby the top left part of the flake is an edge (glass/2L), while the bottom right a Terrace (1L/2L). From the images in Fig 2D, we find that the triangular edges of the bilayer flake and strong defect sites in the center act as primary nucleation sites, as shown in red circle in Fig. R10.

Fig. R10. Visualisation of several 1T nucleation sites (red circles) in 2L under 455 nm illumination (Fig adapted from Fig. 2D in manuscript).

Notably, all nucleation sites indicated above occur roughly at the same point in time with no significant delay irrespective of whether the structure is a terrace or an edge. Further, once nucleated, the phase transformation proceeds through that phase, pushing a (more diffuse) phase boundary along until the entire flake is converted. This behaviour is markedly different than the flake examined under 730 nm illumination, where the 1L terrace nucleated first along the terrace, rather than in its corner. These results suggest that photoexcitation provides

sufficient energy to overcome the difference in nucleation activation energy barriers between edges and terraces and that it prefers nucleation along exposed ('pointy') features of the flake.

2. Thickness dependence

To explore how the observed photo-redox process occurs for thicker samples of more than 2L, we conducted experiment on thicker flakes:

- At 600 nm, we explored a terraced structure with increasingly thicker layers (Section 2-Fig. R5).

We observed nucleation of the 2L with a single wavefront moving from the bottom side of the image across the whole layer, independent to the numbers of layers. This suggests a layer-by-layer approach between successively thicker layers. Importantly, this highlights that even thick layer structures can be converted more rapidly with light exposure in such structure, as the mechanism remains the same, that is, the activation barrier toward nucleation is lowered across the surfaces, helping to move the phase front along.

- At 530 nm, we carried out measurements on a terraced structure with a large 3-4L flake as shown Fig. R11.

As before, the 1T phase nucleates at corner of the 3-4L flake (right hand side in the Fig. R11) and move through this nucleation point across the flake. Interestingly, after about 84 min, we observe distinct intensity regions deep inside the flake, labelled 'a' (orange) and 'b' (purple), which connect the 2H and converted 1T phase. These intermediate region likely correspond layered structures in which 1 or 2 of the 4L structure exhibit 1T character. We highlight that for a large 2L structure illuminated at 600 nm no such phase terracing was observed.

Fig. R11. Snapshot of the phase transition in MoS₂ visualized at 530 nm. (Scale bar = 5 μm)

These results outline that the photo-redox phase transition still applies in thicker flakes but proceeds via a substantially more complex mechanism that depends on the terracing of the structure. This effect may be used to vertically inscribe heterogenous 1T/2H structures, but more research needs to be conducted to explore the optimised parameters for this process. Critically, it highlights that the effect of illumination is not just confined to the surface, but proceeds also via a bulk effect.

Generalisation to thicker systems: To enable rapid conversion of thick flakes in simpler way, photo-redox pathways alone are not sufficient. Instead, we show in our manuscript how redox matching can be utilised to ensure such samples can be converted. This has been explained in the part 2-section 6.

Section 4. Reversibility of the Reaction.

Fig. R12. Ex-situ Raman and PL analysis for monolayer MoS₂. **A.** PL of as exfoliated and photo-redox phase patterned monolayer 1T MoS₂. **B.** PL of photo-redox phase patterned and thermally annealed monolayer 1T MoS₂. **C.** Raman spectra of as exfoliated (1H, bottom) and photo-redox phase patterned 1T MoS₂ (top). **D.** Raman spectra of annealed 1T (photo-redox phase patterned and annealed sample) (red, top), and laser-irradiated thick 1T MoS₂ sample (pink, bottom).

The 1T MoS₂ phase can be converted back to 2H phase by both thermal annealing and laser irradiation^{8,9}. In Fig. R12, the photo-redox patterned monolayer is reversible and converts back to 2H phase when thermal annealing (in Ar/H₂ atmosphere, 300 °C, 3 hours). This was confirmed by PL restoration (B), and Raman spectra (D, top). Here, the E_{2g} peak gets narrower and blue-shifted as the phase restores back to 2H phase by annealing.

The reversibility was also confirmed in thick exfoliated sample (more than 10 layers) using strong laser irradiation. We irradiated a thick 1T sample using a confocal Raman setup (~10 min, 532 nm, 500 mW, focused on diffraction-limited spot (NA=0.85)). The Raman spectra from the thick 1T MoS₂ return to their 2H phase, as the bottom graph of D (pink line). This shows that the 1T phase MoS₂ made by either conventional process (48h n-BuLi treatment) and photo-redox engineering process is reversible.

Section 5. Detailed examination of Raman Data.

1. Analysis on A_{1g} and E_{2g} peaks

In Raman spectra, both A_{1g} and E_{2g} peaks are dampened after phase transition to 1T. This has been demonstrated in calculation and experimental works²⁻⁴. We also observe both peaks are dampened as phase transition to 1T as in Fig. R13, with respect to the n-BuLi treatment duration.

Fig. R13. Raman spectra of monolayer MoS_2 with dependent to the n-BuLi treatment time

In literatures, the E_{2g} peak is more dampened than the A_{1g} peaks during the phase transition to 1T^{2-4,35}. This has been attributed to the metastable nature of 1T MoS_2 in air³⁶. Furthermore, 1T MoS_2 synthesized from the bulk powder in aqueous suspension exhibits dramatically damped E_{2g} and A_{1g} peaks^{3,4}, which rapidly arise again at freshly prepared film with water and air exposure³ (Fig R14 D).

Fig. R14. Raman spectra with dependent to the washing steps. A. Raman signature in two different region. The outer-ring region changes to 1T phase. (scale bar = 5 μm) **B.** n-BuLi treatment followed by n-Hexane washing. **C.** Sample in A followed by Raman measurement and further washing using deionized water, Acetone, and IPA. **D.** Raman signature of 1T MoS₂ suspension and film, data extracted from reference papers^{3,4}.

We examined this washing effect in bulk MoS₂. In Fig. R14, we treated a very thick flake with n-BuLi for 10 hours and washed the sample mildly with n-Hexane before we measured steady-state Raman spectra. Here, we observe clear wave front of two phases similar to our observation in the main text and in literature²¹ (optical image in Fig. R14 A). The Raman spectrum of 1T phase (outer ring) also shows damped and red-shifted signals of E_{2g}, and A_{1g} peak compared to an as-exfoliated sample in the 2H phase. The Raman signal of lithiated MoS₂ in 1T phase (Fig. R14 B) shows lower E_{2g}, A_{1g} peak intensities with higher J-peak intensities (even higher J-peaks ratio than fresh-restacked film in D⁴). However, as we further washed the sample with deionized water (DI), Acetone, and isopropanol (IPA), both E_{2g}, A_{1g} peak gets higher while the J-peak intensity decreases. The Raman spectra from fully-washed sample are similar to 2-h old restacked film in the literature (red curve in Fig R14 D)⁴.

Fig. R15. Ex-situ Raman analysis for monolayer MoS₂ treated with n-BuLi with different time. A. A-exciton PL (around 660 nm) intensity. **B.** Raman E_{2g} peak position. All sample has been washed using deionized water, Acetone, and IPA. Dashed lines are a guide to the eye.

For tracking the dynamics of *ex-situ* Raman E_{2g} mode comparing with PL spectra, we conducted *ex-situ* PL and Raman study using exfoliated monolayer in Fig. R15. A Fig. R15 A shows the *ex-situ* dynamics of A-exciton PL intensity and Fig. R15 B shows the E_{2g} peak position with n-BuLi treatment duration. We made exfoliated monolayer MoS_2 samples and treated with n-BuLi with different duration. The PL intensity is dramatically quenched during the semiconductor (2H) to metal (1T/1T') transition during the first 12 hours. When PL drops significantly, we can observe the red-shift of E_{2g} peak. We note that, the red-shift of 3.8 cm^{-1} is as same as we achieve by photo-redox patterning (3.8 cm^{-1}), showing the robust nature of this feature in assigning the phase transition. This shift is also similar to the literature which showed a red-shift of E_{2g} peak about 2.7 cm^{-1} at phase transition⁵. Therefore, we used the red-shift of E_{2g} peak as a second major indicator for Raman analysis together with the existence of J peaks.

2. Analysis on J-peaks

The emergence of J-peaks is a well-established identification of 1T/1T' phase of TMDs as it has been confirmed after n-BuLi treatment (as in our study)^{8,15,16,37}, during electrochemically lithiation^{26,38} and for chemically grown samples³⁹. However, J-peak shows substantial variation in both position and intensity depending on the sample conditions, which include the starting material form (crystal, powder, chemical vapour deposited), lithiation method, effect of washing steps, air exposure and measurement parameters. The Raman spectra of 1T- MoS_2 exhibits variation in J-peaks in crystal⁸, drop-casted film^{40,41}, CVD (chemical vapor deposition)-grown film^{16,37}, nanostructures^{42,43}, when synthesized from bulk powder^{3,4,44,45}, and mechanically exfoliated sample⁴⁶. Especially, mechanically exfoliated, or CVD-grown mono, and few-layer MoS_2 samples show a very similar Raman signature compared to our data in Fig. 1D (Fig. R16). Moreover, we can observe both E_{2g} , A_{1g} peaks having comparable or higher intensity than J-peaks similar to our data from thicker layer (Fig. R14).

Fig. R16. Literature Raman spectra of 1T phase MoS₂

In Fig. R17, we show the optical and Raman J₂ peak intensity mapping of the photo-redox patterned flake discussed in Fig. 3. Notably, the optical microscope image in Fig. R17 A shows optical contrast of 1T and 2H region is quite different. As the flake becomes metallic, the phase-patterned 1T shows a grey colour while the 2H flake appears somewhat red.

As the J₂ peak is the most intense peak in our 1T monolayer, similar to what was reported for exfoliated samples⁴⁶, we monitored J₂ peak as our representative indication for J peaks. Fig. R17 B shows the peak intensity and position of J₂ peak. As discussed earlier, the peak intensity shows pronounced variations of $\pm 20\%$ (95% confidence interval). even within the same chemical treatment and environment. The direct quantitative comparison of J peak intensities is not a very reliable metric as it originated from the 2 x 1 superlattice structure of distorted 1T phase and the position and intensities shift substantially depending on

environmental factors, as shown in different literature examples^{2,4,36}. Despite this complication, Raman spectroscopy can still be used to identify the metallic phase of MoS₂.

Fig. R17. Raman mapping for photo-redox patterned monolayer MoS₂. **A.** Optical image of patterned flake. **B.** J₂ peak intensity map from the yellow-square region in **A**. Sample has been washed using deionized water, Acetone, and IPA.

From the analysis in Fig. R13-17, we conclude that the dampened Raman E_{2g}, A_{1g} mode with emergence of J-peak can be indicator of 1T phase while it has some variation in its intensity and position, and the shift of E_{2g} peak could be used to monitor the phase transition from 2H to 1T.

Taken together, as there are J-peaks (while even slightly in different intensity) and the identical red-shift of E_{2g} peak in both for dark (48h n-BuLi treated sample without light) and photo-redox patterned sample, we conclude that both sample undergoes the same phase transition process to 1T. It is worth noting that we have monitored our real-time phase transition inside an air-tight cell, but measured Raman in ambient conditions. Furthermore, for safe ex-situ measurements with stable signals, samples were washed out multiple times using DI, Acetone and IPA, for removing the residue of n-BuLi, organic and lithium and lithium oxides on the surface. Without proper washing, the thin layer of MoS₂ gets easily covered with thick organics and lithium oxide layer by air exposure. This process would result in the sample having stronger E_{2g}, A_{1g} peaks and lower J peaks.

Section 6. Redox-based organolithiation agent for novel chemical phase engineering : PAH-Li System.

In Fig. 4, we have provided comprehensive insights into phase engineering by establishing a correlation with *in-situ* optical data and electrochemical potential. We also emphasized the significant role of chemical potential in this process. For MoS₂, the electrochemical potential associated with the phase transition lies around 1.1 V (versus Li/Li⁺), and another lower potential appears around 0.57 V (versus Li/Li⁺), assigned to conversion or amorphization via irreversible decomposition. Consequently, the redox potential of the chemical utilized for the phase transition must be located within these two potential. It aligns well with the spontaneous reaction of n-butyllithium (n-BuLi). The n-BuLi has a redox potential of 1 V (versus Li/Li⁺), initiating phase transition but do not do cause a further conversion process. Based on this understanding and a desire to eliminate pyrophoric n-BuLi, we synthesized novel organolithiation agents to phase engineer TMD materials. We prepared a new reagent by reacting Anthracene or Pyrene with lithium metal in tetrahydrofuran solvent, furnishing a solution of aromatic sigma-radical anions counterbalanced by Li cations.

Table R1 shows chemical formula, structure, and redox potential of different polycyclic aromatic hydrocarbons (PAHs) ^{47–49}. Although the use of different electrolytes, concentration and measurement properties varies the absolute redox potential value, we can identify a trend of decreasing the redox potential as the number of benzene rings decreases. For phase engineering of MoS₂ by lithiation, we have replaced the carbon component ('n-butyl' from 'n-butyllithium') to PAHs having appropriate redox potential, between 1.1 V and 0.57 V (versus Li/Li⁺).

Name	Chemical formula	Chemical structure	Redox potential (Approx. versus Li/Li ⁺)
Tetracene	C ₁₈ H ₁₂		1.44 V
Perylene	C ₂₀ H ₁₂		1.19 ~ 1.35 V
Anthracene	C ₁₄ H ₁₀		0.91 ~ 1.03 V
Pyrene	C ₁₆ H ₁₀		0.73 ~ 0.93 V
Phenanthrene	C ₁₄ H ₁₀		0.31 ~ 0.54 V
Naphthalene	C ₁₀ H ₈		0.26 ~ 0.46 V
Biphenyl	C ₁₂ H ₁₀		0.17 ~ 0.41 V

Table R1. The chemical formula, structure and redox potential (versus Li/Li⁺) of different polycyclic aromatic hydrocarbons (PAHs)

From this table, we selected Naphthalene, Anthracene, Pyrene and Perylene as candidate for studying PAHs-Li system on MoS₂. We used anhydrous tetrahydrofuran (THF) as the organic solvent^{50,51}.

Fig. R18. A. Raman spectra of thick-layered exfoliated MoS₂ treated with PAH-Li systems. **B.** Raman spectra of thick-layered exfoliated MoSe₂ treated with Anthracene-Li systems.

Fig. R18 A shows the Raman spectra measured for thick layer of MoS₂ treated with each PAH-Li. We mechanically exfoliated MoS₂ on glass substrates and treated each PAH-Li agent for 1 hour. We washed the samples using THF and measured Raman using thick-lithiated 1T MoS₂. All sample used here are exfoliated thick samples (more than 4-5 layers by optical contrast at exfoliation). In Fig. R18 A, we can clearly see the 1T signatures of a shifted and broadened E_{2g} peak, with the emergence of J peaks from Pyrene-Li (sky blue), and Anthracene-Li (purple) treated MoS₂. Both PAHs have redox potentials between 1.13 V and 0.57 V and expected to drive phase transition reaction but not further decomposition (conversion, amorphization). Interestingly, compared to n-BuLi which usually takes around 48 hours for the reaction, the reaction using PAH-Li takes less than 1 hour to complete in thick flakes. The use of Naphthalene-Li (green) shows the material undergoing a conversion reaction that has no Raman signature of either 2H, or 1T MoS₂. These results also aligns with literature used Naphthalene-Li and Pyrene-Li in 1,2-Dimethoxyethane (DME) solvent⁴⁸. Conversely, the THF-Anthracene, and Perylene-Li chemicals were unable to drive the reaction.

In Fig. R18 B, we used the Anthracene-Li system on exfoliated, thick MoSe₂. In supplementary Fig.S35-36, MoSe₂ shows two potentials of 0.9 V, and 0.5 V. Therefore, we used anthracene-Li system for 1 hour. The Raman signature (red line in Fig. R 18. B) shows 1T signature as similar to n-BuLi treated MoSe₂ (Fig S34 A in SI). This shows that Anthracene-Li system is working similar to n-BuLi, albeit with faster kinetics.

Method of PAHs-Li: PAHs (ex. Anthracene) were dissolved in anhydrous tetrahydrofuran (THF) solvent to their maximum solubility. Stoichiometric lithium metal (lithium granular) was added to the anthracene-THF solution.

Fig. R19. Photo-redox phase transition using Anthracene-Li system. A. Ex-situ optical analysis for thick MoS₂ flake treated with the anthracene-Li for 600 s without illumination. The dashed line shows clear wave front. **B.** Ex-situ optical analysis for thick MoS₂ flake treated with the anthracene-Li for 10 s with 445 nm illumination. **C.** Raman spectra measured from Sample in **A**. Each spot is designated on **A**. **D.** Raman spectra measured from Sample in **B**. Each spot is designated on **B**. (scale bar for all figure = 5 μm)

We exfoliated a thick MoS₂ (>>10 layers) on glass substrate and placed it in an empty glass vial under ambient conditions before filling it with the anthracene-Li solution. The as-exfoliated each flake is shown in the inset image of C and D. From the optical contrast of each flake, we figure out the thickness of sample used for figure B, D is more thicker than sample for figure A, C. One vial (with sample image in the bottom of Fig C) was allowed to react with the solution for 10 min without any light, while another vial (with sample image in the bottom of Fig D) was exposed to 445 nm LED illumination (SOLIS-1D, power density = 130 mW cm⁻²) for 10 seconds. We washed both samples and monitored the reaction using optical images of each thick flakes *ex-situ*.

In Fig R19 A and B, we observe clear wavefronts as we observed with n-BuLi treated thick-MoS₂²¹. From the wavefront shape, we can roughly calculate a phase front speed for comparison. The wave front speed of flake A (dark) showed ~ 5 nm s⁻¹, while flake B (illuminated) showed a markedly enhanced speed of ~ 5 μm s⁻¹. These front speeds are significantly faster than what we could achieve using n-BuLi with illumination and enable the process even in thick flakes, where the combination of redox matching and above-gap illumination accelerates the reaction significantly. Fig. R19 C, D shows the Raman spectra of each point in Fig. R19 A, B. The inner area (spot 2H₁₋₂ in A and spot 2H₁ in B) remains 2H

phase (red, black curve), while the outer ring region ($1T_{1-2}$ in A, B) changed to 1T phase (blue, cyan curve).

In conclusion, here we synthesize novel redox-based organolithiation agent (PAH-Li) which is easy to use, safe (non-pyrophoric) and perform rapid phase engineering to MoS_2 and $MoSe_2$. This can replace n-BuLi and could make whole process faster and safer and greener. Furthermore, the photo-redox process works with PAH-Li system and accelerates the reaction. When using both photo-redox process with PAH-Li, the whole reaction accelerated 5-6 orders of magnitude than conventional n-BuLi treatment.

Fig. R20. Optical image and Raman spectrum of n-BuLi treated, and Anthracene-Li (AnLi) treated WS₂. **A.** Optical image of n-BuLi treated WS₂. Shrinking core-type phase front is observed. **B.** Raman spectra of 1T WS₂ (red, from outer ring region in A) and 2H WS₂ (black, from inner region in A) **C.** Optical image of AnLi treated WS₂. Two wave fronts are observed. **D.** Raman spectra of 1T WS₂ (red, middle ring region in C) and 2H WS₂ (black, from inner region in C), and converged region (blue, out most region in C) (A, B adapted from Fig. S33 in supplementary information. scale bar for A, C = 20 μm)

We used anthracene-Lithium system to WS₂. We used exfoliated WS₂ on glass substrate and treated AnLi without light in same method as MoS₂ (in Fig. R19 A, C), or MoSe₂ (in Fig. R18 B). When using WS₂ in n-BuLi at the dark, we observed single phase front moving in a shrinking-core fashion inside the flake (Fig. R20 A). In Fig. R20 B, Raman spectra confirmed that the outer region changed to the 1T phase (red), while the core remained 2H (black), with no decomposition products detected.

In Fig. R20 C-D, the use of the anthracene-Lithium driven the reaction further to conversion (decomposition). Optically, in Fig. R20 C, we can identify three concentrically located phases which can be assigned by Raman spectroscopy to 2H in the center, a converted phase (outer ring, blue in D), and an intermediate 1T phase (middle ring region in C and red graph in D). This demonstrates that lithiation of WS₂ follows a two-step process and direct conversion is not the most preferred pathway. While illumination can alter the activation barriers along the different reaction pathways, it is unlikely to change the fundamental mechanism. Mechanistically, this is further supported by our wavelength-dependent studies on MoS₂ (Part 2 - section 2), which highlight that the acceleration of the reaction is a direct consequence of enhance charge production, effectively resulting in an increased driving force/lower activation barrier, rather than a change in the potential response.

References

1. Born, M. *et al.* *Principles of Optics: Electromagnetic Theory of Propagation, Interference and Diffraction of Light*. (Cambridge University Press, 1999). doi:<https://doi.org/10.1017/CBO9781139644181>.
2. Calandra, M. Chemically exfoliated single-layer MoS₂: Stability, lattice dynamics, and catalytic adsorption from first principles. *Phys. Rev. B* **88**, (2013).
3. Yang, D., Sandoval Jimenez, S., Divigalpitiya W. M. R., Irwin, J. C. & Frindt, R. F. Structure of single-molecular-layer MoS₂. *Phys. Rev. B* **43**, 53–56 (1991).
4. Jiménez Sandoval, S., Yang, D., Frindt, R. F. & Irwin, J. C. Raman study and lattice dynamics of single molecular layers of MoS₂. *Phys. Rev. B* **44**, 3955–3962 (1991).
5. Tan, S. J. R. *et al.* Temperature- and Phase-Dependent Phonon Renormalization in 1T'-MoS₂. *ACS Nano* **12**, 5051–5058 (2018).
6. Zhang, J. *et al.* Reversible and selective ion intercalation through the top surface of few-layer MoS₂. *Nat. Commun.* **9**, (2018).
7. Hong, J. *et al.* Exploring atomic defects in molybdenum disulphide monolayers. *Nat. Commun.* **6**, 1–8 (2015).
8. Eda, G. *et al.* Photoluminescence from chemically exfoliated MoS₂. *Nano Lett.* **11**, 5111–5116 (2011).
9. Guo, Y. *et al.* Probing the Dynamics of the Metallic-to-Semiconducting Structural Phase Transformation in MoS₂ Crystals. *Nano Lett.* **15**, 5081–5088 (2015).
10. Goh, K. E. J. *et al.* Quantum transport in two-dimensional WS₂ with high-efficiency carrier injection through indium alloy contacts. *ACS Nano* **14**, 13700–13708 (2020).
11. Ngo, T. D. *et al.* Control of the Schottky Barrier and Contact Resistance at Metal–WSe₂ Interfaces by Polymeric Doping. *Adv. Electron. Mater.* **6**, 1–7 (2020).
12. Pandey, S. *et al.* Transition from direct to Fowler-Nordheim tunneling in chemically reduced graphene oxide film. *Nanoscale* **6**, 3410–3417 (2014).
13. Lee, S. Y. *et al.* Large Work Function Modulation of Monolayer MoS₂ by Ambient Gases. *ACS Nano* **10**, 6100–6107 (2016).
14. Nourbakhsh, A. *et al.* MoS₂ Field-Effect Transistor with Sub-10 nm Channel Length. *Nano Lett.* **16**, 7798–7806 (2016).
15. Kappera, R. *et al.* Phase-engineered low-resistance contacts for ultrathin MoS₂ transistors. *Nat. Mater.* **13**, 1128–1134 (2014).
16. Kappera, R. *et al.* Metallic 1T phase source/drain electrodes for field effect transistors from chemical vapor deposited MoS₂. *APL Mater.* **2**, (2014).

17. Kim, H. J., Yun, Y. J., Yi, S. N., Chang, S. K. & Ha, D. H. Changes in the Photoluminescence of Monolayer and Bilayer Molybdenum Disulfide during Laser Irradiation. *ACS Omega* **5**, 7903–7909 (2020).
18. Hu, L., Shan, X., Wu, Y., Zhao, J. & Lu, X. Laser thinning and patterning of MoS₂ with layer-by-layer precision. *Sci. Rep.* **7**, 1–9 (2017).
19. Rho, Y. *et al.* Site-Selective Atomic Layer Precision Thinning of MoS₂ via Laser-Assisted Anisotropic Chemical Etching. *ACS Appl. Mater. Interfaces* **11**, 39385–39393 (2019).
20. Kang, S. *et al.* Phase-controllable laser thinning in MoTe₂. *Appl. Surf. Sci.* **563**, (2021).
21. Xiong, F. *et al.* Li Intercalation in MoS₂: In Situ Observation of Its Dynamics and Tuning Optical and Electrical Properties. *Nano Lett.* **15**, 6777–6784 (2015).
22. Li, Y. *et al.* Slow Cooling of High-Energy C Excitons Is Limited by Intervalley-Transfer in Monolayer MoS₂. *Laser Photonics Rev.* **13**, 1–7 (2019).
23. Wang, H. *et al.* Electrochemical tuning of vertically aligned MoS₂ nanofilms and its application in improving hydrogen evolution reaction. *Proc. Natl. Acad. Sci. U. S. A.* **110**, 19701–19706 (2013).
24. Quilty, C. D. *et al.* Ex Situ and Operando XRD and XAS Analysis of MoS₂: A Lithiation Study of Bulk and Nanosheet Materials. *ACS Appl. Energy Mater.* **2**, 7635–7646 (2019).
25. Wan, J. *et al.* In situ investigations of Li-MoS₂ with planar batteries. *Adv. Energy Mater.* **5**, 1–7 (2015).
26. Pondick, J. V. *et al.* Thickness-dependent phase transition kinetics in lithium-intercalated MoS₂. *2D Mater.* **9**, (2022).
27. Ghosh, C. *et al.* Phase evolution and structural modulation during in situ lithiation of MoS₂, WS₂ and graphite in TEM. *Sci. Rep.* **11**, (2021).
28. Xu, Y., Kang, J., Hersam, M. C., Wu, J. & Dravid, V. P. Lithium electrochemistry of WS₂ nanoflakes studied by in-situ TEM. *Microsc. Microanal.* **24**, 1860–1861 (2018).
29. Jin, Q., Liu, N., Chen, B. & Mei, D. Mechanisms of Semiconducting 2H to Metallic 1T Phase Transition in Two-dimensional MoS₂ Nanosheets. *J. Phys. Chem. C* **122**, 28215–28224 (2018).
30. Kaushik, N. *et al.* Schottky barrier heights for Au and Pd contacts to MoS₂. *Appl. Phys. Lett.* **105**, (2014).
31. Chee, S. S. *et al.* Lowering the Schottky Barrier Height by Graphene/Ag Electrodes for High-Mobility MoS₂ Field-Effect Transistors. *Adv. Mater.* **31**, 1–7 (2019).

32. Wang, J. *et al.* High Mobility MoS₂ Transistor with Low Schottky Barrier Contact by Using Atomic Thick h-BN as a Tunneling Layer. *Adv. Mater.* **28**, 8302–8308 (2016).
33. Castellanos-Gomez, A., Quereda, J., Van Der Meulen, H. P., Agraït, N. & Rubio-Bollinger, G. Spatially resolved optical absorption spectroscopy of single- and few-layer MoS₂ by hyperspectral imaging. *Nanotechnology* **27**, (2016).
34. Wang, L. *et al.* Slow cooling and efficient extraction of C-exciton hot carriers in MoS₂ monolayer. *Nat. Commun.* **8**, (2017).
35. Attanayake, N. H. *et al.* Effect of Intercalated Metals on the Electrocatalytic Activity of 1T-MoS₂ for the Hydrogen Evolution Reaction. *ACS Energy Lett.* **3**, 7–13 (2018).
36. Jiao, Y. *et al.* Metallic MoS₂ for High Performance Energy Storage and Energy Conversion. *Small* **14**, (2018).
37. Voiry, D. *et al.* The role of electronic coupling between substrate and 2D MoS₂ nanosheets in electrocatalytic production of hydrogen. *Nat. Mater.* **15**, 1003–1009 (2016).
38. El Garah, M. *et al.* MoS₂ nanosheets via electrochemical lithium-ion intercalation under ambient conditions. *FlatChem* **9**, 33–39 (2018).
39. Yu, Y. *et al.* High phase-purity 1T'-MoS₂- and 1T'-MoSe₂-layered crystals. *Nat. Chem.* **10**, 638–643 (2018).
40. Voiry, D. *et al.* Conducting MoS₂ nanosheets as catalysts for hydrogen evolution reaction. *Nano Lett.* **13**, 6222–6227 (2013).
41. Li, Z. *et al.* Lithiated metallic molybdenum disulfide nanosheets for high-performance lithium–sulfur batteries. *Nat. Energy* **8**, 84–93 (2023).
42. Lukowski, M. A. *et al.* Enhanced hydrogen evolution catalysis from chemically exfoliated metallic MoS₂ nanosheets. *J. Am. Chem. Soc.* **135**, 10274–10277 (2013).
43. Ding, Q. *et al.* Efficient photoelectrochemical hydrogen generation using heterostructures of Si and chemically exfoliated metallic MoS₂. *J. Am. Chem. Soc.* **136**, 8504–8507 (2014).
44. Gupta, U. *et al.* Characterization of few-layer 1T-MoSe₂ and its superior performance in the visible-light induced hydrogen evolution reaction. *APL Mater.* **2**, (2014).
45. Wang, Y., Sofer, Z., Luxa, J. & Pumera, M. Lithium Exfoliated Vanadium Dichalcogenides (VS₂, VSe₂, VTe₂) Exhibit Dramatically Different Properties from Their Bulk Counterparts. *Adv. Mater. Interfaces* **3**, 1–8 (2016).
46. Zhang, X. *et al.* Two-dimensional MoS₂-enabled flexible rectenna for Wi-Fi-band wireless energy harvesting. *Nature* **566**, 368–372 (2019).

47. Huang, Y. *et al.* Chemical prelithiation of Al for use as an ambient air compatible and polysulfide resistant anode for Li-ion/S batteries. *J. Mater. Chem. A* **8**, 18715–18720 (2020).
48. Zhu, X. *et al.* Exfoliation of MoS₂ Nanosheets Enabled by a Redox-Potential-Matched Chemical Lithiation Reaction. *Nano Lett.* **22**, 2956–2963 (2022).
49. Zhang, X. *et al.* An electrode-level prelithiation of SiO anodes with organolithium compounds for lithium-ion batteries. *J. Power Sources* **478**, 229067 (2020).
50. Li, F., Cao, Y., Wu, W., Wang, G. & Qu, D. Prelithiation Bridges the Gap for Developing Next-Generation Lithium-Ion Batteries/Capacitors. *Small Methods* **6**, 1–23 (2022).
51. Wang, G. *et al.* High performance lithium-ion and lithium–sulfur batteries using prelithiated phosphorus/carbon composite anode. *Energy Storage Mater.* **24**, 147–152 (2020).

Reviewer Reports on the First Revision:

Referee #1 (Remarks to the Author):

I appreciate the authors' efforts in the revision, and with the added discussion, the experimental and theoretical details are more convincing. However, as I mentioned in the previous comment, the lack of scientific novelty still makes it under the standard of Nature journal. The authors claimed that the highlights of their work lie in the introduction of a safer and faster chemical route, replacing the conventional hazardous method. However, this replacement does not ensure sufficient scientific impact on the TMD semiconductor community, which is more prone to the application of electronic devices. Therefore, I still cannot recommend publication, and another journal with a more focused topic about synthetic materials or materials engineering should be considered.

Referee #2 (Remarks to the Author):

The authors have improved the manuscript with the extensive additional work and they are to be commended.

As in my initial comments, I think the work is technically strong. However, my concerns over the general impact of this work to a broad readership remain.

The general concept of H to T phase engineering using lithiation is well known and even many aspects of the mechanism for the chemical treatment presented here (that is, the dark reaction) have been studied extensively and are, in general, understood. The idea of improving contacts to H-phase TMDs by converting the contact region to T-phase TMDs is also already reported [for one example, see Cho et al. *Science* 349, 625 (2015)]. The greatest element of novelty here is how illumination can be used to accelerate the lithiation process and the possibilities this enables for patterned phase conversion (note however, that the aforementioned *Science* paper showed such possibilities using optically (laser) induced annealing and vacancy formation).

While interesting and of high technical quality, I am still not convinced the insights of this work meet the bar of sufficiently broad interest or specific advance for Nature. In my opinion, I would recommend Nature Materials or Nature Nanotechnology as more suitable venues for this work. But this is perhaps more a question for the editors to decide.

Minor comment:

In Figure 2B, why does the intensity of the 1L recover from 0.9 after the first 50 minutes? The initial dip is discussed in lines 143–148, but I cannot find a discussion of the subsequent recovery.

Referee #3 (Remarks to the Author):

The authors have addressed all of the raised concerns, both in terms of providing more detailed explanations of several unclear points, and, more importantly, in terms of adding new analyses and a group of novel, much faster-acting lithiation agents. To conclude, in spite of several questions still remaining open, the degrees of novelty, significance, and technical quality are at a level corresponding to the Nature journal. I recommend the manuscript to be accepted.

Author Rebuttals to First Revision:

Referee 2

Minor comment:

In Figure 2B, why does the intensity of the 1L recover from 0.9 after the first 50 minutes? The initial dip is discussed in lines 143–148, but I cannot find a discussion of the subsequent recovery.

Thank you for this question. During long-term experiments (>1 h), necessary to capture the phase transition dynamics under 730 nm illumination, we observed noticeable changes to the intensity at the glass/n-BuLi interface of up to 15% over 12 h. These changes match the observed long-term increase observed in Fig. 2 (B, C) in main text and must therefore be correlated. Given that our light sources are stable, the origin of this behaviour must be related to either a change in the refractive index of the n-BuLi solution, or an increased out-of-focus scattering mechanism. In addition to this slow intensity increase, long-term imaging at 730 nm reveals several 'landing events' of several cluster-like particles. These particles are likely formed in solution reactions involving the highly reactive n-BuLi. We therefore believe that the long-term increase in reflection intensity is most likely associated with out-of-focus scattering, which our wide-field microscope will be sensitive to.

We noted this in the main manuscript (highlighted) and detailed explanation is added in Supplementary Information 1.5.